# Value-Incentivized Preference Optimization: A Unified Approach to Online and Offline RLHF

**Shicong Cen**[*]
CMU

**Jincheng Mei**[†]
Google

**Katayoon Goshvadi**[†]
Google

**Hanjun Dai**[†]
Google

**Tong Yang**[*]
CMU

**Sherry Yang**[†]
Google

**Dale Schuurmans**[†]
Google

**Yuejie Chi**[*]
CMU

**Bo Dai**[†]
Google

## Abstract

Reinforcement learning from human feedback (RLHF) has demonstrated great promise in aligning large language models (LLMs) with human preference. Depending on the availability of preference data, both online and offline RLHF are active areas of investigation. A key bottleneck is understanding how to incorporate uncertainty estimation in the reward function learned from the preference data for RLHF, regardless of how the preference data is collected. While the principles of optimism or pessimism under uncertainty are well-established in standard reinforcement learning (RL), a practically-implementable and theoretically-grounded form amenable to large language models is not yet available, as standard techniques for constructing confidence intervals become intractable under arbitrary policy parameterizations. In this paper, we introduce a unified approach to online and offline RLHF — value-incentivized preference optimization (VPO) — which regularizes the maximum-likelihood estimate of the reward function with the corresponding value function, modulated by a *sign* to indicate whether the optimism or pessimism is chosen. VPO also directly optimizes the policy with implicit reward modeling, and therefore shares a simpler RLHF pipeline similar to direct preference optimization. Theoretical guarantees of VPO are provided for both online and offline settings, matching the rates of their standard RL counterparts. Moreover, experiments on text summarization, dialogue, and standard benchmarks verify the practicality and effectiveness of VPO.

## 1 Introduction

Fine-tuning large language models (LLMs) by *reinforcement learning from human feedback* (RLHF) (Ziegler et al., 2019) has been shown to significantly improve the helpfulness, truthfulness and controllability of LLMs, as illustrated by InstructGPT (Ouyang et al., 2022) and many follow-ups. Roughly speaking, there are two critical components of RLHF: (1) *reward modeling*, which maps human preference rankings into a quantitative reward function that can guide policy improvement; and (2) *RL fine-tuning*, which seeks to adjust LLM output to align with human preferences by leveraging the learned reward function, i.e., increasing the probability of preferred answers and decreasing the probability of unfavored answers.

Evidently, the curation of preference data is instrumental in the performance of RLHF, which is commonly modeled as pairwise comparisons from a Bradley-Terry ranking model (Bradley and Terry, 1952). In particular, given a query $x$, human annotators choose a preferred answer from two candidate answers $y_1$ and $y_2$ generated by an LLM. Despite the simple form, collecting large-scale and high-quality preference data can be expensive and time-consuming. Depending on the availability of preference data, two paradigms of RLHF are considered: (1) *offline* RLHF, where only a pre-collected preference dataset is available, possibly generated from a pre-trained LLM after supervised fine-tuning (SFT); and (2) *online* RLHF, where additional preference data can be collected adaptively

---

[*]Carnegie Mellon University; emails: `{shicongc,tongyang,yuejiec}@andrew.cmu.edu`.
[†]Google Research; emails: `{jcmei,kgoshvadi,hadai,sherryy,schuurmans,bodai}@google.com`.

to improve alignment. While initial work on RLHF focused on the offline setting, the online setting has also begun to receive considerable attention, as even a small amount of additional preference data has been shown to greatly boost performance.

There has been significant work on the theoretical underpinnings of RLHF that seeks to uncover algorithmic improvements. Notably, while the original RLHF pipeline decouples reward modeling from RL fine-tuning, direct preference optimization (DPO) (Rafailov et al., 2023) integrates these as a single step in the *offline* setting, leveraging a closed-form solution for the optimal policy in the RL fine-tuning phase. This has led to a welcome simplification of the RLHF pipeline, allowing direct optimization of the policy (i.e., the LLM) from preference data.

Nevertheless, significant challenges remain in RLHF, particularly concerning how to incorporate estimates of reward *uncertainty* in direct preference optimization when parameterizing policies with large-scale neural networks — such as LLMs — in a theoretically and practically effective manner. In standard reinforcement learning (RL), managing uncertainty when an agent interacts with an environment is a critical aspect in achieving near-optimal performance (Sutton and Barto, 2018), when using methods that range from policy-based (Schulman et al., 2017; Xiao et al., 2021), value-based (Mnih et al., 2015; Kumar et al., 2020), and actor-critic methods (Mnih et al., 2016). One dominant approach in the bandit setting, for example, is to construct confidence intervals of the reward estimates, then acting according to the upper and lower confidence bounds — following the principles of optimism and pessimism in the online and offline settings respectively (Lattimore and Szepesvári, 2020; Lai et al., 1985; Rashidinejad et al., 2022).

Despite the fact that uncertainty estimation is even more critical in RLHF, due to the coarse nature of preference data, effective implementations of theoretically justified optimistic and pessimistic principles have yet to be developed in the RLHF literature. For example, existing online preference alignment methods, such as Nash-MD (Munos et al., 2023) and OAIF (Guo et al., 2024), do not incorporate exploration; similarly, pessimism is also not implemented in offline preference alignment methods, such as DPO (Rafailov et al., 2023) and IPO (Azar et al., 2024). A key reason for these omissions is that it is extremely difficult to construct confidence intervals for arbitrary neural networks (Gawlikowski et al., 2021), let alone LLMs. Since optimism for online exploration and pessimism for offline RL both require uncertainty estimation, and given the difficulty of conducting uncertainty estimation for large-scale neural networks, a natural and important question arises:

*Can we implement the optimistic/pessimistic principles under uncertainty in a practically efficient manner for online/offline preference alignment in LLMs while retaining theoretical guarantees?*

## 1.1 OUR CONTRIBUTIONS

In this paper, we provide affirmative answer to the question. Our major contributions are as follows.

**(i)** We propose value-incentivized preference optimization (VPO) for both online and offline RLHF, a unified algorithmic framework that *directly optimizes the LLM policy* with the optimistic/pessimistic principles under uncertainty. Avoiding explicit uncertainty estimation, VPO regularizes maximum likelihood estimation of the reward function toward (resp. against) responses that lead to the highest value in the online (resp. offline) setting, hence implementing optimism (resp. pessimism). Theoretical regret guarantees of VPO are developed for both online and offline RLHF, matching their corresponding rates in the standard RL literature with explicit uncertainty estimation.

**(ii)** In addition, VPO reveals the critical role of reward calibration, where the shift ambiguity of the reward model inherent in the Bradley-Terry model (Bradley and Terry, 1952) can be exploited to implement additional behavior regularization (Pal et al., 2024; Ethayarajh et al., 2024) via centering the reward model with respect to a *calibration policy*. This allows VPO to provide a theoretical foundation for popular conservative offline RL methods (e.g., (Kumar et al., 2020)), as well as regularized RLHF methods (e.g., DPOP (Pal et al., 2024)).

**(iii)** VPO admits a practically-implementable form suitable for RLHF on LLMs, and more generally, deep-learning architectures. We conduct extensive experimental studies using `TL;DR` and `ARC-Challenge` tasks as well as standard benchmarks AlpacaEval 2.0 and MT-Bench in online and offline settings with optimistic and pessimistic bias, respectively. The results demonstrate improved empirical performance.

## 1.2 RELATED WORK

**RLHF.** Since the introduction of the original RLHF framework, there have been many proposed simplifications of the preference alignment procedure and attempts to improve performance, including but not limited to SLiC (Zhao et al., 2023), GSHF (Xiong et al., 2023), DPO (Rafailov et al., 2023), and its variants, such as Nash-MD (Munos et al., 2023), IPO (Azar et al., 2024), OAIF (Guo et al., 2024), SPO (Swamy et al., 2024), SPIN (Chen et al., 2024), WIND (Yang et al., 2024), GPO (Tang et al., 2024), SimPO (Meng et al., 2024), WPO (Zhou et al., 2024), and DPOP (Pal et al., 2024). These methods can roughly be grouped into online and offline variants, depending on whether preference data is collected before training (offline) or by using the current policy during training (online).

In offline preference alignment, identity preference optimization (IPO, (Azar et al., 2024)) argues that it is problematic to use the Bradley-Terry model in DPO to convert pairwise preferences into pointwise reward values, and proposes an alternative objective function to bypass the use of the Bradley-Terry model. DPO-Positive (DPOP, (Pal et al., 2024)) observes a failure mode of DPO that the standard DPO loss can reduce the model's likelihood on preferred answers, and proposes to add a regularization term to the DPO objective to avoid such a failure mode. On the other hand, online AI feedback (OAIF, (Guo et al., 2024)) proposes an online version of DPO, where online preference data from LLM annotators is used to evaluate and update the current LLM policy in an iterative manner. Iterative reasoning preference optimization (Iterative RPO, Pang et al. (2024)) proposes to add an additional negative log-likelihood term in the DPO loss to improve performances on reasoning tasks. Finally, Chang et al. (2024) proposes to reuse the offline preference data via reset. The proposed value-incentivized regularization can be integrated with other improvements to DPO, e.g., SimPO (Meng et al., 2024) and WPO (Zhou et al., 2024), in a straightforward manner.

**Uncertainty estimation in RL.** The principles of optimism and pessimism are typically implemented via constructing confidence intervals or posterior sampling, which have been demonstrated to be provably efficient in tabular settings (Jin et al., 2018; Shi et al., 2022). Yet, these approaches have had limited success in conjunction with deep learning architectures (Gawlikowski et al., 2021), and many empirical heuristics in turn lack theoretical validation (Kumar et al., 2020). Notwithstanding, alternative regularization schemes are developed for general function approximation settings with theoretical guarantees, such as Bellman-consistent pessimism for offline RL (Xie et al., 2021), and reward-biased exploration for online RL (Mete et al., 2021; Liu et al., 2024a) and Markov games (Liu et al., 2024a; Yang et al., 2025). VPO draws inspiration from reward-biased exploration (Kumar and Becker, 1982; Liu et al., 2020; 2024a; Hung et al., 2021; Mete et al., 2021) in the standard online RL literature, but significantly broadens its scope to the offline setting and RLHF for the first time.

**Concurrent work.** While preparing this submission, we discovered several concurrent work that also appeared online around the same time proposing similar regularization techniques as ours to encourage optimism (resp. pessimism) for online (resp. offline) RLHF (Zhang et al., 2024a; Xie et al., 2024; Liu et al., 2024b). Despite slightly different forms, the algorithms studied in Zhang et al. (2024a); Xie et al. (2024); Liu et al. (2024b) can be interpreted as adopting different choices of the *calibration policy* in VPO. In the context of online RLHF, Zhang et al. (2024a) empirically studied a similar algorithm as the proposed online VPO under the contextual bandit formulation of RLHF; Xie et al. (2024) provided finite-time regret analysis of a similar algorithm for the token-level MDP formulation with general function approximation, which extends to general deterministic contextual MDPs as well. In the context of offline RLHF, Liu et al. (2024b) studied a similar algorithm as the proposed offline VPO and provided a sample complexity analysis under the contextual bandit formulation, yet focusing on general function approximation and different assumptions.

## 2 PRELIMINARIES

In RLHF, a language model is described by a policy $\pi$, which generates an answer $y \in \mathcal{Y}$ given prompt $x \in \mathcal{X}$ according to the conditional probability distribution $\pi(\cdot|x)$. The standard RLHF process consists of four stages: supervised fine-tuning (SFT), preference data generation, reward modeling, and RL fine-tuning. In the SFT stage, a language model $\pi_{\text{sft}}$ is obtained by fine-tuning a pre-trained LLM with supervised learning. The remaining stages continue training by leveraging the preference data, which we elaborate below.

**Reward modeling from preference data.** An oracle (e.g., a human labeler or a scoring model) evaluates the quality of two answers $y_1$ and $y_2$ given prompt $x$ and reveals its preference. A widely used approach for modelling the probability of pairwise preferences is the Bradley–Terry model (Bradley and Terry, 1952):

$$\mathbb{P}(y_1 \succ y_2 | x) = \frac{\exp(r^\star(x, y_1))}{\exp(r^\star(x, y_1)) + \exp(r^\star(x, y_2))} = \sigma(r^\star(x, y_1) - r^\star(x, y_2)), \qquad (1)$$

where $y_1 \succ y_2$ indicates that $y_1$ is preferred over $y_2$, $r^\star : \mathcal{X} \times \mathcal{Y} \to \mathbb{R}$ is the ground truth reward function, and $\sigma : \mathbb{R} \to (0, 1)$ is the logistic function. A preference data sample is denoted by a tuple $(x, y_+, y_-)$, where $y_+$ (resp. $y_-$) is the preferred (resp. unpreferred) answer in the comparison.

Given a preference dataset $\mathcal{D} = \{(x^i, y_+^i, y_-^i)\}$ composed of independent samples, the reward function $r$ can be estimated by maximum likelihood estimation (MLE):

$$r_{\mathsf{MLE}} = \arg\min_r \ \ell(r, \mathcal{D}), \qquad (2)$$

where $\ell(r, \mathcal{D})$ is the negative log-likelihood of $\mathcal{D}$, given as $\ell(r, \mathcal{D}) :=$ $-\sum_{(x^i, y_+^i, y_-^i) \in \mathcal{D}} \log \sigma(r(x^i, y_+^i) - r(x^i, y_-^i))$.

**RL fine-tuning.** Given a reward model $r$, we seek to fine-tune the policy $\pi$ to achieve an ideal balance between the expected reward and its distance from an initial policy $\pi_{\text{ref}}$, which is typically the same as $\pi_{\text{sft}}$. This is achieved by maximizing the KL-regularized value function $J(r, \pi)$, defined as

$$J(r, \pi) = \mathbb{E}_{x \sim \rho, y \sim \pi(\cdot|x)} \big[ r(x, y) \big] - \beta \ \mathbb{E}_{x \sim \rho} \big[ \mathsf{KL}\big(\pi(\cdot|x) \,\|\, \pi_{\text{ref}}(\cdot|x)\big) \big], \qquad (3)$$

where $\mathsf{KL}\big(\pi_1 \,\|\, \pi_2\big)$ is the KL divergence from $\pi_1$ to $\pi_2$, and $\beta > 0$ is a regularization parameter. Consequently, the RL fine-tuned policy $\pi_r$ with respect to the reward $r$ satisfies $\pi_r := \arg\max_\pi J(r, \pi)$, which admits a closed-form solution (Rafailov et al., 2023), i.e.,

$$\forall (x \times y) \in \mathcal{X} \times \mathcal{Y} : \qquad \pi_r(y|x) = \frac{\pi_{\text{ref}}(y|x) \exp(r(x, y)/\beta)}{Z(r, x)}. \qquad (4)$$

Here, $Z(r, x)$ is a normalization factor given by $Z(r, x) = \sum_{y' \in \mathcal{Y}} \pi_{\text{ref}}(y'|x) \exp(r(x, y')/\beta)$.

**Direct preference optimization.** The closed-form solution (4) allows us to write the reward function $r$ in turn as

$$r(x, y) = \beta(\log \pi_r(y|x) - \log \pi_{\text{ref}}(y|x) + \log Z(r, x)). \qquad (5)$$

Plugging the above equation into the reward MLE (2), we obtain the seminal formulation of direct preference optimization (DPO) over the policy space (Rafailov et al., 2023),

$$\pi_{\mathsf{DPO}} = \arg\min_\pi \ - \sum_{(x^i, y_+^i, y_-^i) \in \mathcal{D}} \log \sigma \left( \beta \left( \log \frac{\pi(y_+^i|x)}{\pi_{\text{ref}}(y_+^i|x)} - \log \frac{\pi(y_-^i|x)}{\pi_{\text{ref}}(y_-^i|x)} \right) \right), \qquad (6)$$

which avoids explicitly learning the reward model.

## 3 VALUE-INCENTIVIZED PREFERENCE OPTIMIZATION

A major caveat of the standard RLHF framework concerns the lack of accounting for reward uncertainty, which is known to be indispensable in the success of standard RL paradigms in both online and offline settings (Cesa-Bianchi et al., 2017; Rashidinejad et al., 2022). This motivates us to investigate a principled mechanism that be easily integrated into the RLHF pipeline, while bypassing the difficulties of explicit uncertainty estimation in LLMs.

### 3.1 GENERAL FRAMEWORK

In view of the sub-optimality of naive MLE for reward estimation (Cesa-Bianchi et al., 2017; Rashidinejad et al., 2022), and motivated by the effectiveness of reward-biased MLE in online RL (Kumar and Becker, 1982; Liu et al., 2020; 2024a), we propose to regularize the reward estimate via

$$J^\star(r) = \max_\pi J(r, \pi), \qquad (7)$$

which measures the resulting value function for the given reward if one acts according to its optimal policy. However, in RLHF, by the definition (1), the reward function $r^\star$ is only identifiable up to a prompt-dependent global shift. Specifically, letting $r_1(x, y) = r_2(x, y) + c(x)$ be two reward functions that only differ by a prompt-dependent shift $c(x)$, we have $r_1(x, y_1) - r_1(x, y_2) = r_2(x, y_1) - r_2(x, y_2)$, which leads to $J^\star(r_1) = J^\star(r_2) + \mathbb{E}_{x \sim \rho}[c(x)]$. To resolve this challenge, we introduce the following equivalent class of reward functions for the Bradley-Terry model to eliminate the shift ambiguity, which also has the calibration effect of centering the reward function while offering a regularization mechanism to incorporate additional policy preferences.

**Assumption 1** *We assume that $r^\star \in \mathcal{R}$, where*

$$\mathcal{R} = \left\{ r : \mathop{\mathbb{E}}_{x \sim \rho, y \sim \pi_{cal}(\cdot|x)} [r(x, y)] = 0. \right\}. \tag{8}$$

*Here, $\rho$ is the prompt distribution and $\pi_{cal}$ is a fixed calibration distribution independent of the algorithm.*

We remark that the number 0 in the condition can be replaced with arbitrary choice of constant, without affecting derivation of our proposed algorithm. This is due to the Bradley-Terry model being invariant to the global shift in the reward function. Therefore, the condition in Assumption 1 only serves the purpose of notational simplicity and does not put a restrict on the reward model in practice.

The proposed regularized MLE of the Bradley-Terry model (2) appends a bias term to the negative likelihood

$$r_{\text{VPO}} = \arg \min_{r \in \mathcal{R}} \{ \ell(r, \mathcal{D}) - \text{sign} \cdot \alpha \cdot J^\star(r) \}, \tag{9}$$

incentivizing the algorithm to favor (resp. avoid) reward models with higher value $J^\star(r)$ in the online (resp. offline) setting. Here, $\alpha > 0$ is a constant controlling the strength of regularization, and $\text{sign}$ is set to $1$ in the online setting and $-1$ in the offline setting.

At first glance, the objective function for VPO (9) does not immediately imply a computationally-efficient algorithm due to the presence of $J^\star(r)$. However, by exploiting the same closed-form solution for the optimal policy given the reward in (4), and the reward representation inferred from the policy via (5), we can explicitly express $J^\star(r)$ as

$$
\begin{aligned}
J^\star(r) &= \mathop{\mathbb{E}}_{x \sim \rho, y \sim \pi_r(\cdot|x)} [r(x, y) - \beta(\log \pi_r(y|x) - \log \pi_{\text{ref}}(y|x))] \\
&= \mathop{\mathbb{E}}_{x \sim \rho, y \sim \pi_r(\cdot|x)} [\log Z(r, x)] \\
&= \mathop{\mathbb{E}}_{x \sim \rho, y \sim \pi_{\text{cal}}(\cdot|x)} [\log Z(r, x)] \\
&= \mathop{\mathbb{E}}_{x \sim \rho, y \sim \pi_{\text{cal}}(\cdot|x)} [r(x, y) - \beta(\log \pi_r(y|x) - \log \pi_{\text{ref}}(y|x))] \\
&= -\beta \mathop{\mathbb{E}}_{x \sim \rho, y \sim \pi_{\text{cal}}(\cdot|x)} [\log \pi_r(y|x) - \log \pi_{\text{ref}}(y|x)],
\end{aligned} \tag{10}
$$

where the second step follows because the bracketed term is independent of $y$ (c.f. (4)) and the last step follows from (8) whenever $r \in \mathcal{R}$. Given this key ingredient, we can then rewrite (9) to directly optimize the LLM policy, in a flavor similar to DPO, as

$$
\begin{aligned}
\pi_{\text{VPO}} &= \mathop{\arg\min}_{\pi_r: r \in \mathcal{R}} \{ \ell(r, \mathcal{D}) - \text{sign} \cdot \alpha \cdot J^\star(r) \} \\
&= \mathop{\arg\min}_{\pi_r: r \in \mathcal{R}} \left\{ -\sum_{(x^i, y_+^i, y_-^i) \in \mathcal{D}} \log \sigma \left( \beta \log \frac{\pi_r(y_+^i|x^i)}{\pi_{\text{ref}}(y_+^i|x^i)} - \beta \log \frac{\pi_r(y_-^i|x^i)}{\pi_{\text{ref}}(y_-^i|x^i)} \right) \right. \\
&\qquad\qquad \left. + \text{sign} \cdot \alpha\beta \mathop{\mathbb{E}}_{x \sim \rho, y \sim \pi_{\text{cal}}(\cdot|x)} [\log \pi_r(y|x) - \log \pi_{\text{ref}}(y|x)] \right\} \\
&= \mathop{\arg\min}_{\pi} \left\{ -\sum_{(x^i, y_+^i, y_-^i) \in \mathcal{D}} \log \sigma \left( \beta \log \frac{\pi(y_+^i|x^i)}{\pi_{\text{ref}}(y_+^i|x^i)} - \beta \log \frac{\pi(y_-^i|x^i)}{\pi_{\text{ref}}(y_-^i|x^i)} \right) \right. \\
&\qquad\qquad \left. + \text{sign} \cdot \alpha\beta \mathop{\mathbb{E}}_{x \sim \rho, y \sim \pi_{\text{cal}}(\cdot|x)} [\log \pi(y|x) - \log \pi_{\text{ref}}(y|x)] \right\},
\end{aligned} \tag{11}
$$

where we drop the constraint on $r \in \mathcal{R}$, since for any policy $\pi$ there exists $r \in \mathcal{R}$ such that $\pi = \pi_r$.

---

**Algorithm 1** VPO for online RLHF

---

**initialization:** $\pi^{(0)}$.

**for** $t = 0, 1, 2, \cdots$ **do**

    Sample $x^{(t)} \sim \rho$, $y_1^{(t)}, y_2^{(t)} \sim \pi^{(t)}(\cdot | x^{(t)})$.

    Obtain the preference between $(x^{(t)}, y_1^{(t)})$ and $(x^{(t)}, y_2^{(t)})$ from some oracle. Denote the comparison outcome by $(x^{(t)}, y_+^{(t)}, y_-^{(t)})$.

    Update policy $\pi$ as
$$\pi^{(t+1)} = \arg\min_{\pi} \Big\{ - \sum_{s=1}^{t} \log \sigma \Big( \beta \log \frac{\pi(y_+^{(s)}|x^{(s)})}{\pi_{\mathrm{ref}}(y_+^{(s)}|x^{(s)})} - \beta \log \frac{\pi(y_-^{(s)}|x^{(s)})}{\pi_{\mathrm{ref}}(y_-^{(s)}|x^{(s)})} \Big)$$
$$+ \alpha\beta \mathop{\mathbb{E}}_{x \sim \rho, y \sim \pi_{\mathrm{cal}}(\cdot | x)} [\log \pi(y|x) - \log \pi_{\mathrm{ref}}(y|x)] \Big\}. \tag{14}$$

**end**

---

Observing that the reference policy $\pi_{\mathrm{ref}}(y|x)$ in the last term of (11) $\mathop{\mathbb{E}}_{x \sim \rho, y \sim \pi_{\mathrm{cal}}(\cdot|x)} [\log \pi(y|x) - \log \pi_{\mathrm{ref}}(y|x)]$ does not impact the optimization solution, we can change it to $\mathop{\mathbb{E}}_{x \sim \rho, y \sim \pi_{\mathrm{cal}}(\cdot|x)} [\log \pi(y|x) - \log \pi_{\mathrm{cal}}(y|x)] \Big\} = - \mathop{\mathbb{E}}_{x \sim \rho} \big[ \mathsf{KL}\big(\pi_{\mathrm{cal}}(\cdot|x) \,\|\, \pi(\cdot|x)\big) \big]$, which amounts to adding a KL regularization to the original DPO, and offers an interesting interpretation as pushing $\pi$ against/towards $\pi_{\mathrm{cal}}$ in the online/offline settings respectively, unveiling the role of reward calibration in RLHF.

In what follows, we elaborate the development of VPO in both the online and offline settings with corresponding theoretical guarantees under linear function approximation.

## 3.2 ONLINE RLHF: ALGORITHM AND THEORY

The online RLHF procedure extends training by performing reward learning and policy learning iteratively, with a growing preference dataset collected by using the current policy. We use $\pi^{(t)}$ to denote the policy used in the $t$-th iteration, where the superscript $^{(t)}$ indicates iteration $t$ in the online setting. The $t$-th iteration of VPO for online RLHF consists of the following steps:

1. **New preference data generation.** We sample a new prompt $x^{(t)} \sim \rho$ and two answers $y_1^{(t)}, y_2^{(t)} \sim \pi^{(t)}(\cdot|x^{(t)})$, query the preference oracle and append $(x^{(t)}, y_+^{(t)}, y_-^{(t)})$ to the preference dataset.

2. **Reward learning.** We train a reward model with preference data $\mathcal{D}^{(t)} := \{(x^{(s)}, y_+^{(s)}, y_-^{(s)})\}_{s=1}^{t}$ by minimizing the regularized negative log-likelihood, i.e.,
$$r^{(t+1)} = \arg\min_{r \in \mathcal{R}} \{\ell(r, \mathcal{D}^{(t)}) - \alpha \cdot J^\star(r)\}. \tag{12}$$

3. **Policy learning.** This step trains the policy by solving the RL fine-tuning problem:
$$\pi^{(t+1)} = \arg\max_{\pi} J(r^{(t+1)}, \pi). \tag{13}$$

We summarize the detailed procedure in Algorithm 1.

**Theoretical analysis.** Encouragingly, VPO admits appealing theoretical guarantees under function approximation. For simplicity, we restrict attention to linear approximation of the reward model.

**Assumption 2 (Linear Reward)** *We parameterize the reward model by*
$$r_\theta(x, y) = \langle \phi(x, y), \theta \rangle, \quad \forall (x, y) \in \mathcal{X} \times \mathcal{Y}, \tag{15}$$
*where $\phi : \mathcal{X} \times \mathcal{Y} \to \mathbb{R}^d$ is a fixed feature mapping and $\theta \in \mathbb{R}^d$ is the parameters. We assume that $\|\phi(x, y)\|_2 \leq 1$ for all $(x, y) \in \mathcal{X} \times \mathcal{Y}$, and that $r^\star(x, y) = \langle \phi(x, y), \theta^\star \rangle$ for some $\theta^\star$.*

Under Assumption 1 and 2, it is sufficient to focus on $\theta \in \Theta$ where
$$\Theta = \Big\{ \theta \in \mathbb{R}^d : \mathop{\mathbb{E}}_{x \sim \rho, y \sim \pi_{\mathrm{cal}}(\cdot|x)} [\langle \phi(x, y), \theta \rangle] = 0 \Big\}. \tag{16}$$

The next theorem demonstrates that Algorithm 1 achieves $\widetilde{\mathcal{O}}(\sqrt{T})$ cumulative regret under mild assumptions. The proof is provided in Appendix B. The proof logic follows from that of (Liu et al., 2024a).

---

**Algorithm 2** VPO for offline RLHF

---

**input:** offline preference data $\mathcal{D}$ of size $N$.

Get policy $\widehat{\pi}$ by optimizing

$$\widehat{\pi} = \arg\min_{\pi} \left\{ -\sum_{i=1}^{N} \log \sigma \left( \beta \log \frac{\pi(y_+^i|x^i)}{\pi_{\text{ref}}(y_+^i|x^i)} - \beta \log \frac{\pi(y_-^i|x^i)}{\pi_{\text{ref}}(y_-^i|x^i)} \right) \right.$$
$$\left. - \alpha\beta \underset{x\sim\rho, y\sim\pi_{\text{cal}}(\cdot|x)}{\mathbb{E}} [\log \pi(y|x) - \log \pi_{\text{ref}}(y|x)] \right\}.$$

---

**Theorem 1** *Under Assumptions 1 and 2, let $r_{\theta^{(t)}} \in \Theta$ denote the corresponding reward model for $\pi^{(t)}$. Assume that $\|\theta^\star\|_2 \leq C$ and $\|\theta^{(t)}\|_2 \leq C, \forall t \geq 0$ for some $C > 0$. Then with probability $1 - \delta$ we have*

$$\text{Regret} := \sum_{t=1}^{T} \left[ J^\star(r^\star) - J(r^\star, \pi^{(t)}) \right] \leq \widetilde{\mathcal{O}}(\exp(2C + C/\beta)\sqrt{\kappa dT}),$$

*with $\alpha = \dfrac{1}{\exp(2C + C/\beta)} \sqrt{\dfrac{T}{\kappa \min\{d \log T, T\}}}$ and $\kappa = \sup_{x,y} \dfrac{\pi_{cal}(y|x)}{\pi_{ref}(y|x)}$.*

Theorem 1 shows that VPO achieves the same $\widetilde{O}(\sqrt{T})$ regret for online RLHF as its counterparts in standard contextual bandits with scalar rewards and using UCB for exploration (Lattimore and Szepesvári, 2020).

**Remark 1** *The analysis naturally extends to allowing mini-batch samples of size $M$ in every iteration, yielding an improved regret bound scaled by $1/\sqrt{M}$ and $\alpha$ scaled by $\sqrt{M}$.*

### 3.3 Offline RLHF: algorithm and theory

In offline RLHF, a fixed offline preference dataset is collected $\mathcal{D} := \{x^i, y_+^i, y_-^i\}_{i=1}^{N}$, where $x^i \sim \rho$, $y^i \sim \pi_{\text{b}}(\cdot|x)$ are sampled from a behavior policy $\pi_{\text{b}}$, such as $\pi_{\text{sft}}$ from SFT. The proposed VPO for offline RLHF consists of one pass through the reward and policy learning phases, i.e.,

$$\widehat{r} = \arg\min_{r\in\mathcal{R}} \{\ell(r, \mathcal{D}) + \alpha \cdot J^\star(r)\} \qquad \text{and} \qquad \widehat{\pi} = \arg\max_{\pi} J(\widehat{r}, \pi), \tag{17}$$

which discourages over-optimization of the reward function given the limited offline preference data. In the same vein as deriving (14), and by leveraging (10), we obtain the direct policy update rule:

$$\widehat{\pi} = \arg\min_{\pi} \left\{ -\sum_{i=1}^{N} \log \sigma \left( \beta \log \frac{\pi(y_+^i|x^i)}{\pi_{\text{ref}}(y_+^i|x^i)} - \beta \log \frac{\pi(y_-^i|x^i)}{\pi_{\text{ref}}(y_-^i|x^i)} \right) \right.$$
$$\left. - \alpha\beta \underset{x\sim\rho, y\sim\pi_{\text{cal}}(\cdot|x)}{\mathbb{E}} [\log \pi(y|x) - \log \pi_{\text{ref}}(y|x)] \right\}. \tag{18}$$

We summarize the detailed procedure in Algorithm 2. When $\pi_{\text{cal}}$ is set to $\pi_{\text{ref}}$, the regularization term becomes the KL divergence between $\pi$ and $\pi_{\text{ref}}$, which is reminiscent of a popular choice in offline RL practice (Kumar et al., 2020). Another heuristic choice is to set $\pi_{\text{cal}}$ to the marginalized positive answer distribution from the dataset, *i.e.*, $(x, y_+) \sim \mathcal{D}$, which leads to a similar objective in (Pal et al., 2024).

**Saddle-point characterization and pessimism.** We first illustrate that VPO indeed executes the principle of pessimism in a complementary manner to the standard approach of pessimism, which finds a policy that maximizes the worst-case value function over a confidence set. In particular, this strategy (Uehara and Sun, 2021) obtains a policy by solving

$$\widehat{\pi}_{\text{LCB}} = \arg\max_{\pi} \min_{r\in\mathcal{R}_\delta} J(r, \pi) \tag{19}$$

where the confidence set $\mathcal{R}_\delta$ is typically set to $\{r : \ell(r, \mathcal{D}) \leq \ell(r_{\text{MLE}}, \mathcal{D}) + \delta\}$ or $\{r : \text{dist}(r, r_{\text{MLE}}) \leq \delta\}$ for some $\delta > 0$ and s distance measure dist. Turning to VPO, note that by (17) we have

$$\widehat{r} = \arg\min_{r} \{\ell(r, \mathcal{D}) + \alpha J^\star(r)\} = \arg\min_{r} \max_{\pi} \{\ell(r, \mathcal{D}) + \alpha J(r, \pi)\}. \tag{20}$$

Since $\ell(r, \mathcal{D}) + \alpha J(r, \pi)$ is strongly concave over $\pi$, and convex over $r$, it allows us to formulate $(\widehat{r}, \widehat{\pi})$ as a saddle point in the following lemma. The proof is given in Appendix C.1.

**Lemma 1** $(\widehat{r}, \widehat{\pi})$ *is a saddle point of the objective* $\ell(r, \mathcal{D}) + \alpha J(r, \pi)$, *i.e., for any* $(r', \pi')$, *we have*

$$\begin{cases} \ell(\widehat{r}, \mathcal{D}) + \alpha J(\widehat{r}, \widehat{\pi}) \leq \ell(r', \mathcal{D}) + \alpha J(r', \widehat{\pi}) \\ \ell(\widehat{r}, \mathcal{D}) + \alpha J(\widehat{r}, \widehat{\pi}) \geq \ell(\widehat{r}, \mathcal{D}) + \alpha J(\widehat{r}, \pi') \end{cases}.$$

As such, the policy obtained by VPO can be equivalently written as

$$\widehat{\pi} \in \arg\max_{\pi} \min_{r} \left\{ J(r, \pi) + \frac{1}{\alpha} \ell(r, \mathcal{D}) \right\} = \arg\max_{\pi} \min_{r \in \mathcal{R}_{\delta(\pi, \alpha)}} J(r, \pi), \tag{21}$$

where $\mathcal{R}_{\delta(\pi, \alpha)}$ is the constraint set $\{r : \ell(r, \mathcal{D}) \leq \ell(r_{\mathsf{MLE}}, \mathcal{D}) + \delta(\pi, \alpha)\}$ such that the constrained optimization problem $\min_{r \in \mathcal{R}_{\delta(\pi, \alpha)}} J(r, \pi)$ is equivalent to the regularized problem $\min_{r} \{ J(r, \pi) + \frac{1}{\alpha} \ell(r, \mathcal{D}) \}$. In view of the similarity between the formulations (19) and (21), we conclude that VPO implements the pessimism principle (19) in an oblivious manner without explicitly estimating the uncertainty level, justifying popular practice as a valid approach to pessimism (Kumar et al., 2020).

**Theoretical analysis.** The next theorem establishes the sub-optimality gap of VPO with linear function approximation under mild assumptions. The proof is given in Appendix C.

**Theorem 2** *Under Assumptions 1 and 2, let* $\widehat{\theta} \in \Theta$ *denote the corresponding reward model for* $\widehat{\pi}$. *Assume that* $\|\theta^{\star}\|_2 \leq C$ *and* $\|\widehat{\theta}\|_2 \leq C$ *for some* $C > 0$. *Let* $\alpha = \sqrt{N}$ *and* $\delta \in (0, 1)$. *With probability* $1 - \delta$, *we have*

$$J^{\star}(r^{\star}) - J(r^{\star}, \widehat{\pi}) \leq \mathcal{O}\left( \frac{C_1}{\sqrt{N}} \cdot \Big\| \mathop{\mathbb{E}}_{\substack{x \sim \rho, \\ y \sim \pi^{\star}(\cdot|x)}} [\phi(x, y)] \Big\|_{(\Sigma_{\mathcal{D}} + \lambda I)^{-1}} + \frac{C_2}{\sqrt{N}} \right),$$

*where* $\Sigma_{\mathcal{D}} = \frac{1}{N} \sum_{i=1}^{N} (\phi(x^i, y_+^i) - \phi(x^i, y_-^i))(\phi(x^i, y_+^i) - \phi(x^i, y_-^i))^{\top}$ *is the feature sample covariance matrix,* $\lambda = 1/N$, $C_1 = \exp(C)\left( \sqrt{d + \log(1/\delta)} + \kappa_{\mathcal{D}} \right) + C$ *and* $C_2 = \exp(C)\kappa_{\mathcal{D}}^2 + C\kappa_{\mathcal{D}} + 1$. *Here,*

$$\kappa_{\mathcal{D}} = \Big\| \mathop{\mathbb{E}}_{\substack{x \sim \rho, \\ y \sim \widehat{\pi}(\cdot|x)}} [\phi(x, y)] - \mathop{\mathbb{E}}_{\substack{x \sim \rho, \\ y \sim \pi_{cal}(\cdot|x)}} [\phi(x, y)] \Big\|_{(\Sigma_{\mathcal{D}} + \lambda I)^{-1}} \leq 4(\lambda_{\min}(\Sigma_{\mathcal{D}}) + \lambda)^{-1}.$$

Theorem 2 establishes that VPO achieves the same rate of $\widetilde{\mathcal{O}}(1/\sqrt{N})$ as standard offline RL, as long as the offline dataset $\mathcal{D}$ has sufficient coverage. We remark that $\Big\| \mathop{\mathbb{E}}_{\substack{x \sim \rho, \\ y \sim \pi^{\star}(\cdot|x)}} [\phi(x, y)] \Big\|_{(\Sigma_{\mathcal{D}} + \lambda I)^{-1}}$ is reminiscent of the standard single-policy concentratability coefficient in offline RL, which measures the distribution shift between the offline dataset and the optimal policy (Zhu et al., 2023).

**Remark 2** *Recently, Rafailov et al. (2024); Zhong et al. (2024) offered an interpretation of DPO using the token-level Markov Decision Process (MDP), aiming at reconciling the gap between the practical fine-tuning of LLMs at the token level and the theoretical formulation of DPO at the sentence level. Fortunately, VPO can be interpreted using the same setup without introducing further algorithmic modifications. We provide the detailed discussion in Appendix A.*

## 4 EXPERIMENTS

In this section, we evaluate the pessimistic/optimistic VPO for LLMs in offline and online setting, respectively. In both settings, the proposed VPO demonstrates strong performances over the baselines. We provide additional experiments on synthetic bandits in Appendix D.3.

### 4.1 OFFLINE SETTING

In this setting, we focus on `ARC-Challenge` task (Clark et al., 2018), which consists of multiple-choices questions across various science subjects, each with a ground truth answer. To construct the preference pairs for training, we start with $1,119$ examples in the training set and generate three comparison pairs with each incorrect answer, resulting in a total of $3,357$ preference training data. We use `ARC-Challenge` test set which contains $1,172$ questions to test algorithms performances.

We evaluate pessimistic VPO and compare its performance to several offline RLHF baselines (DPO (Rafailov et al., 2023) and IPO (Azar et al., 2024)) on several LLMs, including LLAMA2-7B-CHAT, LLAMA2-13B-CHAT (Touvron et al., 2023) and FLAN-T5-XL (Chung et al., 2022). We emphasize that our goal is to evaluate the RLHF algorithm designs for LLMs, rather than pushing

LLM towards state-of-the-art performance. For fair comparison, we keep all the experiment settings and prompts the same for every RLHF algorithm. We did not apply any additional chain-of-thought reasoning to avoid compounding factors affecting the RLHF performances. We tuned the hyperparameters for VPO and the baselines on the validation set to achieve their best performances. For detailed hyperparameters setup and prompting strategy, please refer to Appendix D.

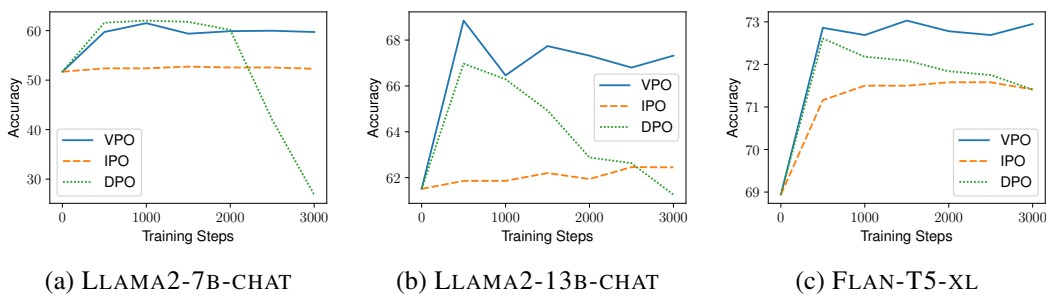

(a) LLAMA2-7B-CHAT   (b) LLAMA2-13B-CHAT   (c) FLAN-T5-XL

Figure 1: The accuracy of the LLAMA2-7B-CHAT, LLAMA2-13B-CHAT and FLAN-T5-XL policies trained by VPO and other baselines (DPO and IPO) on ARC-challenge, respectively. The proposed pessimistic VPO performs consistently strong, and avoids over-optimization.

The performances are illustrated in Figure 1. As we can see, the proposed VPO method demonstrates significantly better performance compared to IPO. Another important observation is that the proposed VPO method is more robust to over-optimization (Gao et al., 2023) compared to DPO. As DPO training continues, its performance declines. In contrast, VPO consistently maintains the performances, avoiding the over-optimization issue and justifying the implicit robustness of pessimism as we revealed in (20).

## 4.2 ONLINE SETTING

For online setting, we conduct two distinct experimental setups. The first, referred to as **Buffer**, adopts the experimental setup in Online AI Feedback (OAIF) (Guo et al., 2024), with an additional buffer used to sample the data for the exploration part of the VPO loss. The second setup, referred to as **Iterative**, adopts the experimental setup in (Zhang et al., 2024a), relying on an online iterative training framework.

**Buffer.** In these set of experiments, we adopt OAIF experimental setup (Guo et al., 2024) where the preference data is gathered by online sampling from the policy and labeled through online feedback. We also introduce a buffer that stores the labeled preferences and is used to sample the data for the exploration term in the VPO loss. We adopt PALM2-XXS language model (Anil et al., 2023) as policy, initialized by supervised finetuning, denoted as SFT model. We exploit another PALM2-XS model as the LLM annotator to provide online feedback. We evaluate the performance of optimistic VPO and compare its performance to Online DPO (Guo et al., 2024). We choose TL;DR task (Stiennon et al., 2020) and extract its prompts for the input of preference data. Similar to (Guo et al., 2024), we use *Detailed 0-shot* prompt from Lee et al. (2023). The prompts we used and how we get preference scores are detailed in Appendix D. We emphasize our algorithm is agnostic to human or AI feedback.

As a sanity check, we track the win rate of VPO and Online DPO against the SFT baseline on TL;DR during training in Figure 2a. For ablation purpose, we vary the exploration weight $\alpha = \{0.01, 0.1\}$ in the optimistic VPO. One significant observation is that although all the online RLHF algorithms follow the increase trend, the win-rate against SFT of the optimistic VPO has larger oscillation, comparing to Online DPO. And the oscillation reduces, with $\alpha$ diminishing. Our conjecture is that this behavior is encouraged by the optimistic term in VPO, for collecting more unexplored data, which may delay the learning due to the diversity in data. However, as the learning proceeds, the proposed VPO outperforms the competitors, because of the coverage of the collected data. To demonstrate the advantages of optimistic VPO in online setting more directly, we evaluate the win/tie/loss rate against Online DPO head-to-head, as shown in Figure 2b. This clearly shows that the optimistic VPO achieves better performances with larger exploration preference, and thus, consolidates our conclusion that *i)*, the simple value-incentivized term makes the exploration practical without uncerntainty estimation; and *ii)*, exploration is potentially beneficial for better model.

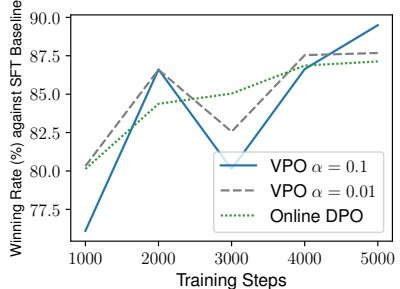 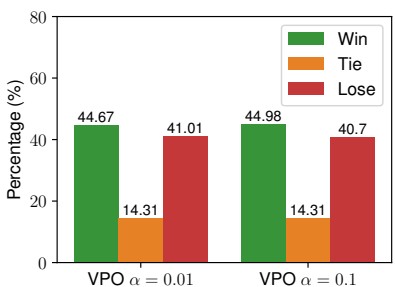

Figure 2(a): Win rate of VPO and Online DPO against the SFT baseline on `TL;DR` task.

Figure 2(b): Win/tie/loss rate of VPO with different exploration rate $\alpha = \{0.01, 0.1\}$, directly against Online DPO.

**Iterative.** We further evaluate the performance of VPO on standard benchmarks AlpacaEval 2.0 (Dubois et al., 2024b;a) and MT-Bench (Zheng et al., 2024), using similar experimental setup in recent literature (Zhang et al., 2024b; Wu et al., 2024). We use UltraFeedback[1] (Cui et al., 2023) as our training dataset which contains around 61k preference pairs of single-turn conversations. We split the 61k prompts into four chunks and follow an iterative training approach. We choose Zephyr-7B-SFT[2] as our LLM model. We follow the best hyperparameters setup found in Zhang et al. (2024b) and compare the results of VPO to DPO reported therein. We first conduct a single iteration of standard DPO training on the first portion of the training data, referred to as Zephyr-7B-DPO in 1. We then perform 3 iterations of VPO, each iteration on a different data portion, while using online AI feedback provided by PairRM (Jiang et al., 2023) in between. Further details of our experiments are explained in Appendix D.

The evaluation results are summarized in Table 1, deferred to Appendix D, filling out the baselines based on Zhang et al. (2024b). As could been seen, VPO significantly improves the performance of the base model Zephyr-7B-SFT by **14.52**, achieving the highest length-controlled (LC) Win Rate on the AlpacaEval 2.0 benchmark, beating DPO. This result, 22.53, is competitive to much larger models such as much Yi-34B-Chat, 27.19, and Llama-3-70B-Instruct, 33.17. Additionally, VPO shows significant improvement on MT-Bench compared to the base model Zephyr-7B-SFT with the increase of **2.32** while beating DPO. We further report the results of other iterative post-training algorithms, such as SPIN (Chen et al., 2024), DNO (Rosset et al., 2024), and SPPO (Wu et al., 2024) and show that even though VPO is trained on a weak base model, it achieves comparable results to these baselines. Granular views on a radar chart can also be found in Appendix D.

## 5 CONCLUSION AND DISCUSSION

In this work, we develop a unified approach to achieving principled optimism and pessimism in online and offline RLHF, which enables a practical computation scheme by incorporating uncertainty estimation implicitly within reward-biased maximum likelihood estimation. Theoretical analysis indicates that the proposed methods mirror the guarantees of their standard RL counterparts, which is furthermore corroborated by numerical results. Important future directions include investigating adaptive rules for selecting $\alpha$ without prior information and more refined analysis on the choice of $\pi_{\text{cal}}$. This work also hints at a general methodology of designing practical algorithms with principled optimism/pessimism under more general RL setups.

## ACKNOWLEDGEMENT

The work of S. Cen, T. Yang and Y. Chi is supported in part by the grants ONR N00014-19-1-2404, NSF DMS-2134080, CCF-2106778 and CNS-2148212. S. Cen is also gratefully supported by Wei Shen and Xuehong Zhang Presidential Fellowship and JP Morgan AI Research Fellowship.

---

[1]https://huggingface.co/datasets/HuggingFaceH4/ultrafeedback_binarized
[2]https://huggingface.co/HuggingFaceH4/mistral-7b-sft-beta

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

## A  TOKEN-LEVEL VPO

**Token-level MDP and preference modeling.** Recall that in LLMs, the prompt $x$ can be broken into a sequence of tokens, e.g., $x = (x_0, \ldots, x_m)$, from a fixed discrete vocabulary $\mathcal{A}$. We define the token-level MDP as a tuple $\mathcal{M} = (\mathcal{S}, \mathcal{A}, P, r^\star, H)$, where $H$ is the horizon length, i.e., the longest possible number of tokens in a sentence. The state space $\mathcal{S}$ consists of all the possible token combinations of length $H$, and the transition kernel is deterministically defined as follows.

1. The initial state is defined by the prompt $x$ as $s_0 = \{x_0, \cdots, x_m\}$;
2. Given the response $y = \{y_0, \cdots, y_{i-1}\}$ up to the $i$-th token, the state at step $i$ is defined as $s_i = \{x_0, \cdots, x_m, y_0, \cdots, y_{i-1}\}$;
3. Upon an action of the LLM for the next token $a_i = y_i$, the next state at the token-level MDP deterministically becomes $s_{i+1} = (s_i, a_i) = (x_0, \cdots, x_m, y_0, \cdots, y_i)$.

We assume that the last token of a sentence, the EOS token, is absorbing, such that the token-level MDP stays in the corresponding state as soon as the last action is the EOS token. With slight abuse of notation from earlier sections, the reward function $r^\star(s, a)$ defines the ground truth reward at state $s$ upon action $a$.

Given a pair of trajectories $\tau_1 = \{s_0, a_0^1, \ldots, s_{H-1}^1, a_{H-1}^1, s_H^1\}$ and $\tau_2 = \{s_0, a_0^2, \ldots, s_{H-1}^2, a_{H-1}^2, s_H^2\}$, the corresponding Bradley-Terry preference model (Bradley and Terry, 1952) is

$$\mathbb{P}(\tau_1 \succ \tau_2) = \frac{\exp\left(\sum_{i=0}^{H-1} r^\star(s_i^1, a_i^1)\right)}{\exp\left(\sum_{i=0}^{H-1} r^\star(s_i^1, a_i^1)\right) + \exp\left(\sum_{i=0}^{H-1} r^\star(s_i^2, a_i^2)\right)}$$

$$= \sigma \left( \sum_{i=0}^{H-1} r^\star(s_i^1, a_i^1) - \sum_{i=0}^{H-1} r^\star(s_i^2, a_i^2) \right),$$

A preference data sample is denoted by a tuple $(x, \tau_+, \tau_-)$, where $\tau_+$ (resp. $\tau_-$) is the preferred (resp. unpreferred) answer in the comparison. Given a preference dataset $\mathcal{D}$ composed of independent samples, The negative log-likelihood can be defined as

$$\ell(r, \mathcal{D}) := - \sum_{(\tau_+, \tau_-) \in \mathcal{D}} \log \sigma \left( \sum_{i=0}^{H-1} r(s_i^+, a_i^+) - \sum_{i=0}^{H-1} r(s_i^-, a_i^-) \right), \tag{22}$$

where $(s_i^+, a_i^+)$ (resp. $(s_i^-, a_i^-)$) are the state-action pairs in the trajectory $\tau_+$ (resp. $\tau_-$).

**Token-level RL fine-tuning.** Let the entropy of policy $\pi$ under initial state distribution $s_0 \sim \rho$ be defined as

$$\mathcal{H}(\rho, \pi) := - \mathbb{E}_{\substack{s_0 \sim \rho, \\ a_i \sim \pi(\cdot|s_i)}} \left[ \sum_{i=0}^{H-1} \log \pi(a_i|s_i) \right],$$

and $\mathcal{H}(s, \pi)$ be the entropy when the initial state $s_0 = s$. Given a reward function $r$, we define the KL-constrained RL objective against the reference policy $\pi_{\text{ref}}$ as (Rafailov et al., 2024):

$$\pi_r := \operatorname*{argmax}_\pi J(r, \pi) := \mathbb{E}_{\substack{s_0 \sim \rho, \\ a_i \sim \pi(\cdot|s_i)}} \left[ \sum_{i=0}^{H-1} \underbrace{(r(s_i, a_i) + \beta \log \pi_{\text{ref}}(a_i|s_i))}_{r_\beta(s_i, a_i)} \right] + \beta \mathcal{H}(\rho, \pi), \tag{23}$$

where we denote $r_\beta : \mathcal{S} \times \mathcal{A} \to \mathbb{R}$ as follows:

$$\forall (s, a) \in \mathcal{S} \times \mathcal{A} : \quad r_\beta(s, a) := r(s, a) + \beta \log \pi_{\text{ref}}(a|s), \tag{24}$$

which can be seen as the actual token-wise reward function optimized by the LLM.

**Token-level DPO.** The KL-constrained RL objective (23) has a closed-form solution (Nachum et al., 2017; Cen et al., 2022) given by

$$\forall (s, a) \in \mathcal{S} \times \mathcal{A} : \quad \pi_r(a|s) = \exp((Q_\beta^\star(s, a) - V_\beta^\star(s))/\beta), \tag{25}$$

where $V_\beta^\star : \mathcal{S} \to \mathbb{R}$ and $Q_\beta^\star : \mathcal{S} \times \mathcal{A} \to \mathbb{R}$ are the optimal soft value and Q functions, respectively,

$$\forall s \in \mathcal{S} : \quad V_\beta^\star(s) := \mathbb{E}_{a_i \sim \pi_r(\cdot|s_i)} \left[ \sum_{i=0}^{H-1} r_\beta(s_i, a_i) | s_0 = s \right] + \beta \mathcal{H}(s, \pi_r) \tag{26}$$

denote the optimal soft value function w.r.t. the reward function $r_\beta$, and

$$\forall (s, a) \in \mathcal{S} \times \mathcal{A} : \quad Q_\beta^\star(s, a) := r_\beta(s, a) + V_\beta^\star(s'), \tag{27}$$

where $s' = (s, a)$ is the deterministic next state. Plugging (27) and (24) into (25) implies that, for any trajectory $\tau = \{s_0, a_0, \cdots, a_{H-1}, s_H\}$, Rafailov et al. (2024) shows

$$\sum_{i=0}^{H-1} r(s_i, a_i) = \sum_{i=0}^{H-1} \left( Q_\beta^\star(s_i, a_i) - \beta \log \pi_{\text{ref}}(a_i|s_i) - V_\beta^\star(s_{i+1}) \right)$$

$$= Q_\beta^\star(s_0, a_0) - \beta \log \pi_{\text{ref}}(a_0|s_0) + \sum_{i=1}^{H-1} \left( Q_\beta^\star(s_i, a_i) - V_\beta^\star(s_i) - \beta \log \pi_{\text{ref}}(a_i|s_i) \right)$$

$$= V_\beta^\star(s_0) + \beta \sum_{i=0}^{H-1} \log \frac{\pi_r(a_i|s_i)}{\pi_{\text{ref}}(a_i|s_i)}, \tag{28}$$

where the second line uses $V_\beta^\star(s_H) = 0$ at the terminal state. Thus the DPO loss (which is the negative log-likelihood loss) could be written as

$$\mathcal{L}(\pi, \mathcal{D}) = - \sum_{(\tau_+, \tau_-) \in \mathcal{D}} \log \sigma \left( \beta \sum_{i=0}^{H-1} \log \frac{\pi(a_i^+|s_i^+)}{\pi_{\text{ref}}(a_i^+|s_i^+)} - \beta \sum_{i=0}^{H-1} \log \frac{\pi(a_i^-|s_i^-)}{\pi_{\text{ref}}(a_i^-|s_i^-)} \right). \tag{29}$$

**Token-level VPO.** With slight abuse of notation, define

$$J^\star(r) := \max_\pi J(r, \pi), \tag{30}$$

which is used as the bias term in regularizing the reward estimation in VPO. Again, we impose the following assumption to deal with the shift ambiguity issue caused by the Bradley-Terry model:

**Assumption 3** *We assume that $r^\star \in \mathcal{R}$, where*

$$\mathcal{R} = \left\{ r : \mathop{\mathbb{E}}_{\substack{s_0 \sim \rho, \\ a_i \sim \pi_{cal}(\cdot|s_i)}} \sum_{i=0}^{H-1} r(s_i, a_i) = 0 \right\}. \tag{31}$$

*Here, $\rho$ is the prompt distribution and $\pi_{cal}$ is a fixed calibration distribution independent of the algorithm.*

Combining (23) with (26), similar to previous derivations, we have

$$
\begin{aligned}
J^\star(r) &= \mathop{\mathbb{E}}_{s_0 \sim \rho} \left[ V_\beta^\star(s_0) \right] \\
&= \mathop{\mathbb{E}}_{\substack{s_0 \sim \rho, \\ a_i \sim \pi_{\mathrm{cal}}(\cdot|s_i)}} \left[ V_\beta^\star(s_0) \right] \\
&= \mathop{\mathbb{E}}_{\substack{s_0 \sim \rho, \\ a_i \sim \pi_{\mathrm{cal}}(\cdot|s_i)}} \left[ \sum_{i=0}^{H-1} r(s_i, a_i) - \beta \sum_{i=0}^{H-1} \log \frac{\pi_r(a_i|s_i)}{\pi_{\mathrm{ref}}(a_i|s_i)} \right] \\
&= -\beta \mathop{\mathbb{E}}_{\substack{s_0 \sim \rho, \\ a_i \sim \pi_{\mathrm{cal}}(\cdot|s_i)}} \left[ \sum_{i=0}^{H-1} \log \frac{\pi_r(a_i|s_i)}{\pi_{\mathrm{ref}}(a_i|s_i)} \right],
\end{aligned}
\tag{32}
$$

where the penultimate line uses (28), and the last line uses Assumption 3.

Consequently, the token-level VPO can be rewritten as

$$
\begin{aligned}
\pi_{\mathsf{VPO}} = \arg\min_\pi \Bigg\{ &- \sum_{(\tau_+, \tau_-) \in \mathcal{D}} \log \sigma \left( \beta \sum_{i=0}^{H-1} \log \frac{\pi(a_i^+|s_i^+)}{\pi_{\mathrm{ref}}(a_i^-|s_i^-)} - \beta \sum_{i=0}^{H-1} \log \frac{\pi(a_i^-|s_i^-)}{\pi_{\mathrm{ref}}(a_i^-|s_i^-)} \right) \\
&+ \mathsf{sign} \cdot \alpha\beta \mathop{\mathbb{E}}_{\substack{s_0 \sim \rho, \\ a_i \sim \pi_{\mathrm{cal}}(\cdot|s_i)}} \left[ \sum_{i=0}^{H-1} \log \frac{\pi(a_i|s_i)}{\pi_{\mathrm{ref}}(a_i|s_i)} \right] \Bigg\}.
\end{aligned}
\tag{33}
$$

## B  ANALYSIS FOR THE ONLINE SETTING

### B.1  PROOF OF THEOREM 1

For ease of presentation, we assume that $\mathcal{R}$ is finite, i.e., $|\mathcal{R}| < \infty$. The general case can be directly obtained using a covering number argument, which we refer to (Liu et al., 2024a; Jin et al., 2022) for interested readers.

We start by decomposing the regret into two parts:

$$
\begin{aligned}
\mathsf{Regret} &:= \sum_{t=1}^T \left[ J^\star(r^\star) - J(r^\star, \pi^{(t)}) \right] \\
&= \underbrace{\sum_{t=1}^T \left[ J^\star(r^\star) - J^\star(r^{(t)}) \right]}_{\textbf{Term (i)}} + \underbrace{\sum_{t=1}^T \left[ J(r^{(t)}, \pi^{(t)}) - J(r^\star, \pi^{(t)}) \right]}_{\textbf{Term (ii)}}.
\end{aligned}
\tag{34}
$$

**Step 1: bounding term (i).** By the choice of $r^{(t)}$, we have

$$\ell(r^{(t)}, \mathcal{D}^{(t-1)}) - \alpha J^\star(r^{(t)}) \le \ell(r^\star, \mathcal{D}^{(t-1)}) - \alpha J^\star(r^\star). \tag{35}$$

Rearranging terms,

$$J^\star(r^\star) - J^\star(r^{(t)}) \le \frac{1}{\alpha} \left[ \ell(r^\star, \mathcal{D}^{(t-1)}) - \ell(r^{(t)}, \mathcal{D}^{(t-1)}) \right]. \tag{36}$$

The following lemma is adapted from (Liu et al., 2024a, Proposition 5.3), whose proof is deferred to Appendix B.2.

**Lemma 2** *Let $\delta \in (0, 1)$. With probability $1 - \delta$, we have*

$$\ell(r^\star, \mathcal{D}^{(t-1)}) - \ell(r^{(t)}, \mathcal{D}^{(t-1)})$$

$$\leq -2 \sum_{s=1}^{t-1} \mathop{\mathbb{E}}_{\substack{x \sim \rho, \\ (y_1, y_2) \sim \pi^{(s)}(\cdot|x)}} \left[ D_H^2(\mathbb{P}_{r^{(t)}}(\cdot|x, y_1, y_2) \| \mathbb{P}_{r^\star}(\cdot|x, y_1, y_2)) \right] + 2 \log(|\mathcal{R}|/\delta). \quad (37)$$

*Here, $D_H(\cdot\|\cdot)$ is the Hellinger distance, $\mathbb{P}_r(\cdot|x, y_1, y_2)$ denotes the Bernoulli distribution of the comparison result of $(x, y_1)$ and $(x, y_2)$ under reward model $r$.*

Putting the above inequalities together, it holds with probability $1 - \delta$ that

**Term (i)** $\leq -\dfrac{2}{\alpha} \displaystyle\sum_{t=1}^{T} \sum_{s=1}^{t-1} \mathop{\mathbb{E}}_{\substack{x^{(s)} \sim \rho, \\ (y_1^{(s)}, y_2^{(s)}) \sim \pi^{(s)}(\cdot|x^{(s)})}} \left[ D_H^2(\mathbb{P}_{r^{(t)}}(\cdot|x^{(s)}, y_1^{(s)}, y_2^{(s)}) \| \mathbb{P}_{r^\star}(\cdot|x^{(s)}, y_1^{(s)}, y_2^{(s)})) \right]$

$$+ 2\alpha^{-1} T \log(|\mathcal{R}|/\delta). \quad (38)$$

**Step 2: breaking down term (ii) with the elliptical potential lemma.** The linear function approximation form (15) allows us to write

$$\mathop{\mathbb{E}}_{x \sim \rho, y \sim \pi_{r_2}(\cdot|x)} [r_1(x, y) - r^\star(x, y)] = \langle W(r_1), X(r_2) \rangle, \quad (39)$$

where $X, W : \mathcal{R} \to \mathbb{R}^d$ is given by

$$X(r_\theta) = 2C \mathop{\mathbb{E}}_{x \sim \rho, y \sim \pi_{r_\theta}(\cdot|x)} [\phi(x, y)], \qquad W(r_\theta) = \frac{\theta - \theta^\star}{2C}. \quad (40)$$

Let

$$\Sigma_t = \epsilon I + \sum_{s=1}^{t-1} X(r^{(t)}) X(r^{(t)})^\top \quad (41)$$

for some $\epsilon > 0$. We begin by decomposing term (ii) as

$$\textbf{Term (ii)} = \sum_{t=1}^{T} \mathop{\mathbb{E}}_{x \sim \rho, y \sim \pi^{(t)}(\cdot|x)} \left[ r^{(t)}(x, y) - r^\star(x, y) \right]$$

$$= \sum_{t=1}^{T} \left\langle W(r^{(t)}), X(r^{(t)}) \right\rangle$$

$$= \sum_{t=1}^{T} \left\langle W(r^{(t)}), X(r^{(t)}) \right\rangle \mathbf{1}\{\|X(r^{(t)})\|_{\Sigma_t^{-1}} \leq 1\}$$

$$+ \sum_{t=1}^{T} \left\langle W(r^{(t)}), X(r^{(t)}) \right\rangle \mathbf{1}\{\|X(r^{(t)})\|_{\Sigma_t^{-1}} > 1\}, \quad (42)$$

where $\mathbf{1}\{A\}$ is an indicator function of event $A$. To proceed, we recall the elliptical potential lemma for controlling the cumulative sum of $\min\{\|X(r^{(t)})\|_{\Sigma_t^{-1}}^2, 1\}$.

**Lemma 3** ((Abbasi-Yadkori et al., 2011, Lemma 11)) *Let $\{X_t\}$ be a sequence in $\mathbb{R}^d$ and $\Lambda_0 \in \mathbb{R}^{d \times d}$ a positive definite matrix. Define $\Lambda_t = \Lambda_0 + \sum_{s=1}^{t} X_s X_s^\top$. Assume $\|X_t\| \leq L$ for all $t$. It holds that*

$$\sum_{t=1}^{T} \min\{\|X_t\|_{\Lambda_t^{-1}}^2, 1\} \leq 2 \log\left(\frac{det(\Lambda_T)}{det(\Lambda_0)}\right)$$

$$\leq 2(d \log((trace(\Lambda_0) + TL^2)/d) - \log det(\Lambda_0)).$$

Applying the above lemma yields

$$\sum_{t=1}^{T} \min\{\|X(r^{(t)})\|_{\Sigma_t^{-1}}^2, 1\} \leq \min\left\{2d \log\left(\frac{4C^4 T/d + \epsilon}{\epsilon}\right), T\right\} := d(\epsilon). \quad (43)$$

We now control the two terms in (42).

- The first term of (42) can be bounded by

$$\sum_{t=1}^{T} \left\langle W(r^{(t)}), X(r^{(t)}) \right\rangle \mathbf{1}\{\|X(r^{(t)})\|_{\Sigma_t^{-1}} \le 1\}$$

$$\le \sum_{t=1}^{T} \|W(r^{(t)})\|_{\Sigma_t} \|X(r^{(t)})\|_{\Sigma_t^{-1}} \mathbf{1}\{\|X(r^{(t)})\|_{\Sigma_t^{-1}} \le 1\}$$

$$\le \sum_{t=1}^{T} \|W(r^{(t)})\|_{\Sigma_t} \min\left\{ \|X(r^{(t)})\|_{\Sigma_t^{-1}}, 1 \right\}$$

$$= \sum_{t=1}^{T} \left[ \epsilon\|W(r^{(t)})\|_2^2 + \sum_{s=1}^{t-1} \left\langle W(r^{(t)}), X(r^{(s)}) \right\rangle^2 \right]^{1/2} \min\left\{ \|X(r^{(t)})\|_{\Sigma_t^{-1}}^2, 1 \right\}^{1/2}$$

$$\overset{(i)}{\le} \left\{ \sum_{t=1}^{T} \left[ \epsilon\|W(r^{(t)})\|_2^2 + \sum_{s=1}^{t-1} \left\langle W(r^{(t)}), X(r^{(s)}) \right\rangle^2 \right] \right\}^{1/2} \left\{ \sum_{t=1}^{T} \min\left\{ \|X(r^{(t)})\|_{\Sigma_t^{-1}}^2, 1 \right\} \right\}^{1/2}$$

$$\overset{(ii)}{\le} \sqrt{d(\epsilon)} \left\{ \sum_{t=1}^{T} \sum_{s=1}^{t-1} \left\langle W(r^{(t)}), X(r^{(s)}) \right\rangle^2 \right\}^{1/2} + \sqrt{d(\epsilon)\epsilon T}$$

$$\overset{(iii)}{\le} \frac{d(\epsilon)}{2\mu} + \frac{\mu}{2} \sum_{t=1}^{T} \sum_{s=1}^{t-1} \left\langle W(r^{(t)}), X(r^{(s)}) \right\rangle^2 + \sqrt{d(\epsilon)\epsilon T}. \tag{44}$$

  Here, (i) is due to Cauchy–Schwarz inequality, (ii) is due to $\sqrt{a+b} \le \sqrt{a} + \sqrt{b}$ for $\forall a, b \ge 0$, and (iii) results from Young's inequality. We leave the constant $\mu > 0$ to be determined later.

- The second term of (42) can be bounded by

$$\sum_{t=1}^{T} \left\langle W(r^{(t)}), X(r^{(t)}) \right\rangle \mathbf{1}\{\|X(r^{(t)})\|_{\Sigma_t^{-1}} > 1\} \le C \sum_{t=1}^{T} \mathbf{1}\{\|X(r^{(t)})\|_{\Sigma_t^{-1}} > 1\}$$

$$\le C \sum_{t=1}^{T} \min\{\|X(r^{(t)})\|_{\Sigma_t^{-1}}^2, 1\} \le Cd(\epsilon), \tag{45}$$

  where the first inequality follows from $\|X(r^{(t)})\|_2 \le 2C$ and $\|W(r^{(t)})\|_2 \le 1/2$ since $\|\phi(x,y)\|_2 \le 1$.

Putting (42), (44) and (45) together, we arrive at

$$\textbf{Term (ii)} \le \frac{d(\epsilon)}{2\mu} + \frac{\mu}{2} \sum_{t=1}^{T} \sum_{s=1}^{t-1} \left\langle W(r^{(t)}), X(r^{(s)}) \right\rangle^2 + \sqrt{d(\epsilon)\epsilon T} + Cd(\epsilon). \tag{46}$$

**Step 3: continuing bounding term (ii).** It boils down to control $\left\langle W(r^{(t)}), X(r^{(s)}) \right\rangle^2$. We have

$$\left\langle W(r^{(t)}), X(r^{(s)}) \right\rangle = \mathop{\mathbb{E}}_{\substack{x\sim\rho, \\ y\sim\pi^{(s)}(\cdot|x)}} \left[ r^{(t)}(x,y) - r^{\star}(x,y) \right]$$

$$= \mathop{\mathbb{E}}_{\substack{x\sim\rho, \\ y_1\sim\pi^{(s)}(\cdot|x)}} \left[ r^{(t)}(x,y_1) - r^{\star}(x,y_1) \right] - \mathop{\mathbb{E}}_{\substack{x\sim\rho, \\ y_2\sim\pi_{\text{cal}}(\cdot|x)}} \left[ r^{(t)}(x,y_2) - r^{\star}(x,y_2) \right]$$

$$= \mathop{\mathbb{E}}_{\substack{x\sim\rho, \\ y_1\sim\pi^{(s)}(\cdot|x), \\ y_2\sim\pi_{\text{cal}}(\cdot|x)}} \left[ \delta_x(r^{(t)}, r^{\star}, y_1, y_2) \right], \tag{47}$$

where $\delta_x(r_1, r_2, y_1, y_2) := r_1(x, y_1) - r_1(x, y_2) - (r_2(x, y_1) - r_2(x, y_2))$. Therefore,

$$
\begin{aligned}
\left\langle W(r^{(t)}), X(r^{(s)}) \right\rangle^2 &= \mathop{\mathbb{E}}_{\substack{x \sim \rho, \\ y_1 \sim \pi^{(s)}(\cdot|x), \\ y_2 \sim \pi_{\mathrm{cal}}(\cdot|x)}} \left[ \delta_x(r^{(t)}, r^\star, y_1, y_2) \right]^2 \\
&= \mathop{\mathbb{E}}_{\substack{x \sim \rho, \\ y_1 \sim \pi^{(s)}(\cdot|x), \\ y_2 \sim \pi_{\mathrm{cal}}(\cdot|x)}} \left[ \delta_x(r^{(t)}, r^\star, y_1, y_2)^2 \right] - \mathop{\mathrm{Var}}_{\substack{x \sim \rho, \\ y_1 \sim \pi^{(s)}(\cdot|x), \\ y_2 \sim \pi_{\mathrm{cal}}(\cdot|x)}} \left[ \delta_x(r^{(t)}, r^\star, y_1, y_2)^2 \right] \\
&\leq \mathop{\mathbb{E}}_{\substack{x \sim \rho, \\ y_1 \sim \pi^{(s)}(\cdot|x), \\ y_2 \sim \pi_{\mathrm{cal}}(\cdot|x)}} \left[ \delta_x(r^{(t)}, r^\star, y_1, y_2)^2 \right] \\
&\leq \sup_{x,y} \frac{\pi_{\mathrm{cal}}(y|x)}{\pi^{(s)}(y|x)} \cdot \mathop{\mathbb{E}}_{\substack{x \sim \rho, \\ y_1, y_2 \sim \pi^{(s)}(\cdot|x)}} \left[ \delta_x(r^{(t)}, r^\star, y_1, y_2)^2 \right] \\
&\leq \sup_{x,y} \frac{\pi_{\mathrm{ref}}(y|x)}{\pi^{(s)}(y|x)} \cdot \sup_{x,y} \frac{\pi_{\mathrm{cal}}(y|x)}{\pi_{\mathrm{ref}}(y|x)} \cdot \mathop{\mathbb{E}}_{\substack{x \sim \rho, \\ y_1, y_2 \sim \pi^{(s)}(\cdot|x)}} \left[ \delta_x(r^{(t)}, r^\star, y_1, y_2)^2 \right]. \quad (48)
\end{aligned}
$$

Recall from (4) that $\pi^{(s)}(y|x) \propto \pi_{\mathrm{ref}}(y|x) \exp(r^{(s)}(x,y)/\beta)$. It follows that $|\log \pi^{(s)}(y|x) - \log \pi_{\mathrm{ref}}(y|x)| \leq 2\|r^{(s)}(x,\cdot)\|_\infty \leq 2C/\beta$ (see *e.g.*, (Cen et al., 2022, Appendix A.2)), and hence $\sup_{x,y} \frac{\pi_{\mathrm{ref}}(y|x)}{\pi^{(s)}(y|x)} \leq \exp(2C/\beta)$. To proceed, we demonstrate in the following lemma that $\delta^2$ can be upper bounded by the corresponding Hellinger distance, whose proof is deferred to Appendix B.3.

**Lemma 4** *Assume bounded reward* $\|r_1\|_\infty \leq C$, $\|r_2\|_\infty \leq C$. *We have*
$$
\delta_x(r_1, r_2, y_1, y_2)^2 \leq 2(3 + \exp(2C))^2 D_H^2(\mathbb{P}_{r_1}(\cdot|x, y_1, y_2) \,\|\, \mathbb{P}_{r_2}(\cdot|x, y_1, y_2)).
$$

With the above lemma we arrive at
$$
\begin{aligned}
&\left\langle W(r^{(t)}), X(r^{(s)}) \right\rangle^2 \\
&\leq 2(3 + \exp(2C))^2 \exp(2C/\beta)\kappa \cdot \mathop{\mathbb{E}}_{\substack{x \sim \rho, \\ y_1, y_2 \sim \pi^{(s)}(\cdot|x)}} \left[ D_H^2(\mathbb{P}_{r^{(t)}}(\cdot|x, y_1, y_2) \,\|\, \mathbb{P}_{r^\star}(\cdot|x, y_1, y_2)) \right].
\end{aligned}
$$
where we denote $\kappa = \sup_{x,y} \frac{\pi_{\mathrm{cal}}(y|x)}{\pi_{\mathrm{ref}}(y|x)}$. Plugging the above bound into (46), we get

**Term (ii)**

$$
\begin{aligned}
&\leq \frac{d(\epsilon)}{2\mu} + \mu(3 + \exp(2C))^2 \exp(2C/\beta)\kappa \cdot \sum_{t=1}^{T} \sum_{s=1}^{t-1} \mathop{\mathbb{E}}_{\substack{x \sim \rho, \\ y_1, y_2 \sim \pi^{(s)}(\cdot|x)}} \left[ D_H^2(\mathbb{P}_{r^{(t)}}(\cdot|x, y_1, y_2) \,\|\, \mathbb{P}_{r^\star}(\cdot|x, y_1, y_2)) \right] \\
&\quad + 2B\sqrt{d(\epsilon)\epsilon T} + Cd(\epsilon). \quad (49)
\end{aligned}
$$

**Step 4: finishing up.** Combining (34), (38) and (49), with probability $1 - \delta$ we have
$$
\mathsf{Regret} \leq \frac{2T \log(|\mathcal{R}|/\delta)}{\alpha} + \frac{d(\epsilon)}{2\mu} + \sqrt{d(\epsilon)\epsilon T} + Cd(\epsilon) \quad (50)
$$

as long as $\alpha\mu(3 + \exp(2C))^2 \exp(2C/\beta)\kappa \leq 2$. Setting $\alpha \asymp \frac{1}{\exp(2C+C/\beta)}\sqrt{\frac{T}{\kappa d(\epsilon)}}$, $\mu \asymp \frac{1}{\exp(2C+C/\beta)}\sqrt{\frac{d(\epsilon)}{\kappa T}}$, and $\epsilon = 1$, we arrive at
$$
\mathsf{Regret} \leq \widetilde{\mathcal{O}}((\exp(2C + C/\beta))\sqrt{\kappa dT})
$$
as claimed.

### B.2 PROOF OF LEMMA 2

To begin, we have
$$
\ell(r^\star, \mathcal{D}^{(t-1)}) - \ell(r^{(t)}, \mathcal{D}^{(t-1)}) = -\log \frac{\mathbb{P}(\mathcal{D}^{(t-1)}|r^\star)}{\mathbb{P}(\mathcal{D}^{(t-1)}|r^{(t)})} = -\sum_{s=1}^{t-1} X_{r^{(t)}}^s, \quad (51)
$$

where we denote

$$X_r^s = \log \frac{\mathbb{P}_{r^\star}(y_+^{(s)} \succ y_-^{(s)}|x^{(s)})}{\mathbb{P}_r(y_+^{(s)} \succ y_-^{(s)}|x^{(s)})}. \tag{52}$$

To proceed, we recall a useful martingale exponential inequality.

**Lemma 5 ((Zhang, 2023, Theorem 13.2),(Liu et al., 2024a, Lemma D.1))** *Let $\{X_t\}_{t=1}^\infty$ be a sequence of real-valued random variables adapted to filtration $\{\mathcal{F}_t\}_{t=1}^\infty$. It holds with probability $1-\delta$ such that for any $t \geq 1$,*

$$-\sum_{s=1}^t X_s \leq \sum_{s=1}^t \log \mathbb{E}\left[\exp(-X_s)|\mathcal{F}_{s-1}\right] + \log(1/\delta).$$

Applying the above lemma to $\{\frac{1}{2}X_r^t\}_{t=1}^\infty$ along with the filtration $\{\mathcal{F}_t\}_{t=1}^\infty$ with $\mathcal{F}_t$ given by the $\sigma$-algebra of $\{(x^{(s)}, y_+^{(s)}, y_-^{(s)}) : s \leq t\}$, we conclude that it holds with probability $1-\delta$ that

$$-\frac{1}{2}\sum_{s=1}^{t-1} X_r^s \leq \sum_{s=1}^{t-1} \log \mathbb{E}\left[\exp\left\{-\frac{1}{2}X_r^s\right\}\Big|\mathcal{F}_{s-1}\right] + \log(|\mathcal{R}|/\delta)$$

$$\leq \sum_{s=1}^{t-1} \left(\mathbb{E}\left[\exp\left\{-\frac{1}{2}X_r^s\right\}\Big|\mathcal{F}_{s-1}\right] - 1\right) + \log(|\mathcal{R}|/\delta), \tag{53}$$

where the last step results from the inequality $\log(1+x) \leq x$ for all $x \geq -1$. To proceed, note that

$$\mathbb{E}\left[\exp\left\{-\frac{1}{2}X_r^s\right\}\Big|\mathcal{F}_{s-1}\right]$$

$$= \mathbb{E}\left[\sqrt{\frac{\mathbb{P}_r(y_+^{(s)} \succ y_-^{(s)}|x^{(s)})}{\mathbb{P}_{r^\star}(y_+^{(s)} \succ y_-^{(s)}|x^{(s)})}}\,\Big|\,\mathcal{F}_{s-1}\right]$$

$$= \mathop{\mathbb{E}}_{\substack{x^{(s)}\sim\rho, \\ (y_1^{(s)},y_2^{(s)})\sim\pi^{(s)}(\cdot|x^{(s)}), \\ (+,-)\sim\mathbb{P}_{r^\star}}}\left[\sqrt{\frac{\mathbb{P}_r(y_+^{(s)} \succ y_-^{(s)}|x^{(s)})}{\mathbb{P}_{r^\star}(y_+^{(s)} \succ y_-^{(s)}|x^{(s)})}}\right]$$

$$= \mathop{\mathbb{E}}_{\substack{x^{(s)}\sim\rho, \\ (y_1^{(s)},y_2^{(s)})\sim\pi^{(s)}(\cdot|x^{(s)})}}\left[\sum_{(+,-)} \sqrt{\mathbb{P}_r(y_+^{(s)} \succ y_-^{(s)}|x^{(s)}) \cdot \mathbb{P}_{r^\star}(y_+^{(s)} \succ y_-^{(s)}|x^{(s)})}\right]$$

$$= 1 - \frac{1}{2}\mathop{\mathbb{E}}_{\substack{x^{(s)}\sim\rho, \\ (y_1^{(s)},y_2^{(s)})\sim\pi^{(s)}(\cdot|x^{(s)})}}\left[\sum_{(+,-)} \left(\sqrt{\mathbb{P}_r(y_+^{(s)} \succ y_-^{(s)}|x^{(s)})} - \sqrt{\mathbb{P}_{r^\star}(y_+^{(s)} \succ y_-^{(s)}|x^{(s)})}\right)^2\right]$$

$$= 1 - \mathop{\mathbb{E}}_{\substack{x\sim\rho, \\ (y_1,y_2)\sim\pi^{(s)}(\cdot|x)}}\left[D_\mathrm{H}^2(\mathbb{P}_r(\cdot|x,y_1,y_2 \,\|\, \mathbb{P}_{r^\star}(\cdot|x,y_1,y_2)\right],$$

where we denote by $\sum_{(+,-)}$ the summation over different comparison results. Plugging the above equality into (53) completes the proof.

### B.3 PROOF OF LEMMA 4

By the mean value theorem, we have

$$\left|\mathbb{P}_{r_1}(y_1 \succ y_2|x) - \mathbb{P}_{r_2}(y_1 \succ y_2|x)\right| = \left|\sigma(r_1(x,y_1) - r_1(x,y_2)) - \sigma(r_2(x,y_1) - r_2(x,y_2))\right|$$

$$= \left|\delta_x(r_1, r_2, y_1, y_2) \cdot \sigma'(\xi)\right|$$

$$= \left|\delta_x(r_1, r_2, y_1, y_2)\right| \cdot \sigma(\xi)(1 - \sigma(\xi))$$

for some $\xi$ between $r_1(x,y_1) - r_1(x,y_2)$ and $r_2(x,y_1) - r_2(x,y_2)$. Since $|\xi| \leq 2C$, we have

$$\sigma(\xi)(1-\sigma(\xi)) \geq \sigma(2C)(1-\sigma(2C)) \geq \frac{1}{3 + \exp(2C)}. \tag{54}$$

Putting together,

$$\begin{aligned}
\big|\delta_x(r_1, r_2, y_1, y_2)\big| &\leq (3 + \exp(2C))\big|\mathbb{P}_{r_1}(y_1 \succ y_2|x) - \mathbb{P}_{r_2}(y_1 \succ y_2|x)\big| \\
&= (3 + \exp(2C))\mathbf{TV}(\mathbb{P}_{r_1}(\cdot|x, y_1, y_2), \mathbb{P}_{r_2}(\cdot|x, y_1, y_2)) \\
&\leq (3 + \exp(2C))\sqrt{2}D_{\mathrm{H}}(\mathbb{P}_{r_1}(\cdot|x, y_1, y_2) \,\|\, \mathbb{P}_{r_2}(\cdot|x, y_1, y_2)).
\end{aligned}$$

# C   ANALYSIS FOR THE OFFLINE SETTING

## C.1   PROOF OF LEMMA 1

By definition, the objective function $\ell(r, \mathcal{D}) + \alpha J(r, \pi)$ is strongly concave over $\pi$, and convex over $r$. By Danskin's theorem, we have

$$\nabla_r\big(\max_\pi[\ell(\widehat{r}, \mathcal{D}) + \alpha J(\widehat{r}, \pi)]\big) = \nabla_r\big(\ell(\widehat{r}, \mathcal{D}) + \alpha J(\widehat{r}, \widehat{\pi})\big).$$

Therefore, for any $r'$, by convexity of the objective function we have

$$\begin{aligned}
\ell(r', \mathcal{D}) + \alpha J(r', \widehat{\pi}) &\geq \ell(\widehat{r}, \mathcal{D}) + \alpha J(\widehat{r}, \widehat{\pi}) + \big\langle r' - \widehat{r}, \nabla_r\big(\ell(\widehat{r}, \mathcal{D}) + \alpha J(\widehat{r}, \widehat{\pi})\big)\big\rangle \\
&= \ell(\widehat{r}, \mathcal{D}) + \alpha J(\widehat{r}, \widehat{\pi}) + \big\langle r' - \widehat{r}, \nabla_r\big(\max_\pi[\ell(\widehat{r}, \mathcal{D}) + \alpha J(\widehat{r}, \pi)]\big)\big\rangle \\
&\geq \ell(\widehat{r}, \mathcal{D}) + \alpha J(\widehat{r}, \widehat{\pi}).
\end{aligned}$$

The last line is due to the definition of $\widehat{r}$ (c.f. (20)). The other relation, $\ell(\widehat{r}, \mathcal{D}) + \alpha J(\widehat{r}, \widehat{\pi}) \geq \ell(\widehat{r}, \mathcal{D}) + \alpha J(\widehat{r}, \pi')$, follows directly from the definition of $\widehat{\pi}$ (c.f. (17)).

## C.2   PROOF OF THEOREM 2

We decompose the sub-optimality gap of $\widehat{\pi}$ by

$$\begin{aligned}
&J^\star(r^\star) - J(r^\star, \widehat{\pi}) \\
&= \big[J(r^\star, \pi^\star) - J(\widehat{r}, \pi^\star)\big] + \big[J(\widehat{r}, \pi^\star) - J(\widehat{r}, \widehat{\pi})\big] + \big[J(\widehat{r}, \widehat{\pi}) - J(r^\star, \widehat{\pi})\big] \\
&\leq \underbrace{\big[J(r^\star, \pi^\star) - J(\widehat{r}, \pi^\star)\big]}_{\textbf{Term (i)}} + \underbrace{\big[J(\widehat{r}, \widehat{\pi}) - J(r^\star, \widehat{\pi})\big]}_{\textbf{Term (ii)}},
\end{aligned} \tag{55}$$

where the last line is due to $J(\widehat{r}, \pi^\star) \leq J(\widehat{r}, \widehat{\pi})$ according to the definition of $\widehat{\pi}$ (c.f. (17)). We proceed to bound the two terms separately. Here we have written $\widehat{r} = r_{\widehat{\theta}}$ for notational simplicity. In addition, we denote the MLE estimate by $r_{\mathsf{MLE}} = r_{\theta_{\mathsf{MLE}}}$.

By the definition of $J(r, \pi)$ (cf. (3)), it follows that term (i) in (55) can be further decomposed as

$$\begin{aligned}
\textbf{Term (i)} &= \mathop{\mathbb{E}}_{\substack{x \sim \rho, \\ y \sim \pi^\star(\cdot|x)}} [r^\star(x, y) - \widehat{r}(x, y)] \\
&= \mathop{\mathbb{E}}_{\substack{x \sim \rho, \\ y \sim \pi^\star(\cdot|x)}} \Big[\big\langle\phi(x, y), \theta^\star - \widehat{\theta}\big\rangle\Big] \\
&= \underbrace{\mathop{\mathbb{E}}_{\substack{x \sim \rho, \\ y \sim \pi^\star(\cdot|x)}} \Big[\big\langle\phi(x, y), \theta^\star - \theta_{\mathsf{MLE}}\big\rangle\Big]}_{\textbf{Term (ia)}} + \underbrace{\mathop{\mathbb{E}}_{\substack{x \sim \rho, \\ y \sim \pi^\star(\cdot|x)}} \Big[\big\langle\phi(x, y), \theta_{\mathsf{MLE}} - \widehat{\theta}\big\rangle\Big]}_{\textbf{Term (ib)}},
\end{aligned} \tag{56}$$

where $r_{\mathsf{MLE}}(x, y) = \langle\phi(x, y), \theta_{\mathsf{MLE}}\rangle$.

**Step 1: bounding term (ia).** To continue, we recall a useful lemma from (Zhu et al., 2023).

**Lemma 6 ((Zhu et al., 2023, Lemma 3.1))** *For any $\lambda > 0$ and $\delta \in (0, 1)$, with probability at least $1 - \delta$,*

$$\|\theta_{\mathsf{MLE}} - \theta^\star\|_{\Sigma_{\mathcal{D}} + \lambda I} \leq \mathcal{O}\left((3 + \exp(C))\sqrt{\frac{d + \log(1/\delta)}{N}} + \sqrt{\lambda C^2}\right).$$

*In addition, we have*

$$\frac{1}{3 + \exp(C)}\Sigma_{\mathcal{D}} \preceq \frac{1}{N}\nabla_\theta^2\ell(r_\theta, \mathcal{D}) \preceq \frac{1}{4}\Sigma_{\mathcal{D}} \tag{57}$$

*for all $\theta$ such that $\|r_\theta\|_\infty \leq C$.*

The first term of (56) can be bounded with Lemma 6 as

$$\textbf{Term (ia)} \le \|\theta^\star - \theta_{\mathsf{MLE}}\|_{\Sigma_{\mathcal{D}}+\lambda I} \cdot \Big\| \mathbb{E}_{\substack{x\sim\rho,\\ y\sim\pi^\star(\cdot|x)}} [\phi(x,y)] \Big\|_{(\Sigma_{\mathcal{D}}+\lambda I)^{-1}}$$

$$\le \mathcal{O}\Bigg( \Big( (3+\exp(C))\sqrt{\frac{d+\log(1/\delta)}{N}} + \sqrt{\lambda C^2} \Big) \cdot \Big\| \mathbb{E}_{\substack{x\sim\rho,\\ y\sim\pi^\star(\cdot|x)}} [\phi(x,y)] \Big\|_{(\Sigma_{\mathcal{D}}+\lambda I)^{-1}} \Bigg).$$

$$(58)$$

**Step 2: bounding term (ib).** For the second term of (56), recall that
$$\widehat{r} = \arg\min_{r\in\mathcal{R}} \big\{ \ell(r,\mathcal{D}) + \alpha J(r,\widehat{\pi}) \big\},$$

or equivalently

$$\widehat{\theta} = \arg\min_{\theta\in\Theta} \big\{ \ell(r_\theta,\mathcal{D}) + \alpha J(r_\theta,\widehat{\pi}) \big\},$$

and that

$$\theta_{\mathsf{MLE}} = \arg\min_{\theta\in\Theta} \ell(r_\theta,\mathcal{D}).$$

With linear constraint (16), by KKT condition we have

$$\nabla_\theta \ell(\widehat{r},\mathcal{D}) + \alpha \mathbb{E}_{\substack{x\sim\rho,\\ y\sim\widehat{\pi}(\cdot|x)}} [\phi(x,y)] + \lambda_1 \mathbb{E}_{\substack{x\sim\rho,\\ y\sim\pi_{\mathrm{cal}}(\cdot|x)}} [\phi(x,y)] = 0$$

for some $\lambda_1 \in \mathbb{R}$, and

$$\nabla_\theta \ell(r_{\mathsf{MLE}},\mathcal{D}) + \lambda_2 \mathbb{E}_{\substack{x\sim\rho,\\ y\sim\pi_{\mathrm{cal}}(\cdot|x)}} [\phi(x,y)] = 0$$

for some $\lambda_2 \in \mathbb{R}$. By strong monotonicity of $\nabla_\theta \ell$ (cf. (57)), we have

$$\frac{N}{3+\exp(C)} \big\| \widehat{\theta} - \theta_{\mathsf{MLE}} \big\|_{\Sigma_{\mathcal{D}}}^2 \le \big\langle \nabla_\theta \ell(\widehat{r},\mathcal{D}) - \nabla_\theta \ell(r_{\mathsf{MLE}},\mathcal{D}), \widehat{\theta} - \theta_{\mathsf{MLE}} \big\rangle$$

$$= \Big\langle -\alpha \mathbb{E}_{\substack{x\sim\rho,\\ y\sim\widehat{\pi}(\cdot|x)}} [\phi(x,y)] - (\lambda_1 - \lambda_2) \mathbb{E}_{\substack{x\sim\rho,\\ y\sim\pi_{\mathrm{cal}}(\cdot|x)}} [\phi(x,y)], \widehat{\theta} - \theta_{\mathsf{MLE}} \Big\rangle$$

$$= -\alpha \Big\langle \mathbb{E}_{\substack{x\sim\rho,\\ y\sim\widehat{\pi}(\cdot|x)}} [\phi(x,y)] - \mathbb{E}_{\substack{x\sim\rho,\\ y\sim\pi_{\mathrm{cal}}(\cdot|x)}} [\phi(x,y)], \widehat{\theta} - \theta_{\mathsf{MLE}} \Big\rangle$$

$$\le \alpha \Big\| \mathbb{E}_{\substack{x\sim\rho,\\ y\sim\widehat{\pi}(\cdot|x)}} [\phi(x,y)] - \mathbb{E}_{\substack{x\sim\rho,\\ y\sim\pi_{\mathrm{cal}}(\cdot|x)}} [\phi(x,y)] \Big\|_{(\Sigma_{\mathcal{D}}+\lambda I)^{-1}} \big\| \widehat{\theta} - \theta_{\mathsf{MLE}} \big\|_{\Sigma_{\mathcal{D}}+\lambda I}$$

$$\le \alpha \kappa_{\mathcal{D}} \big\| \widehat{\theta} - \theta_{\mathsf{MLE}} \big\|_{\Sigma_{\mathcal{D}}+\lambda I},$$

where we denote

$$\kappa_{\mathcal{D}} = \Big\| \mathbb{E}_{\substack{x\sim\rho,\\ y\sim\widehat{\pi}(\cdot|x)}} [\phi(x,y)] - \mathbb{E}_{\substack{x\sim\rho,\\ y\sim\pi_{\mathrm{cal}}(\cdot|x)}} [\phi(x,y)] \Big\|_{(\Sigma_{\mathcal{D}}+\lambda I)^{-1}}. \tag{59}$$

The penultimate step results from $\widehat{\theta}, \theta_{\mathsf{MLE}} \in \Theta$, which ensures

$$\Big\langle \mathbb{E}_{\substack{x\sim\rho,\\ y\sim\pi_{\mathrm{cal}}(\cdot|x)}} [\phi(x,y)], \widehat{\theta} \Big\rangle = \Big\langle \mathbb{E}_{\substack{x\sim\rho,\\ y\sim\pi_{\mathrm{cal}}(\cdot|x)}} [\phi(x,y)], \theta_{\mathsf{MLE}} \Big\rangle = 0$$

It follows that

$$\frac{N}{3+\exp(C)} \big\| \widehat{\theta} - \theta_{\mathsf{MLE}} \big\|_{\Sigma_{\mathcal{D}}+\lambda I}^2 \le \frac{N}{3+\exp(C)} \big\| \widehat{\theta} - \theta_{\mathsf{MLE}} \big\|_{\Sigma_{\mathcal{D}}}^2 + \frac{N}{3+\exp(C)} \big\| \widehat{\theta} - \theta_{\mathsf{MLE}} \big\|_{\lambda I}^2$$

$$\le \alpha \kappa_{\mathcal{D}} \big\| \widehat{\theta} - \theta_{\mathsf{MLE}} \big\|_{\Sigma_{\mathcal{D}}+\lambda I} + \frac{N\lambda C^2}{3+\exp(C)}.$$

The above inequality allows us to bound

$$\big\| \widehat{\theta} - \theta_{\mathsf{MLE}} \big\|_{\Sigma_{\mathcal{D}}+\lambda I} \le \frac{\alpha(3+\exp(C))}{N} \kappa_{\mathcal{D}} + 2\sqrt{\lambda C^2}. \tag{60}$$

Therefore, the second term of (56) can be bounded as

$$\textbf{Term (ib)} \leq \left\| \widehat{\theta} - \theta_{\mathsf{MLE}} \right\|_{\Sigma_{\mathcal{D}} + \lambda I} \left\| \mathop{\mathbb{E}}_{\substack{x \sim \rho, \\ y \sim \pi^\star(\cdot|x)}} [\phi(x,y)] \right\|_{(\Sigma_{\mathcal{D}} + \lambda I)^{-1}}$$

$$\leq \left( \frac{\alpha(3 + \exp(C))}{N} \kappa_{\mathcal{D}} + 2\sqrt{\lambda C^2} \right) \left\| \mathop{\mathbb{E}}_{\substack{x \sim \rho, \\ y \sim \pi^\star(\cdot|x)}} [\phi(x,y)] \right\|_{(\Sigma_{\mathcal{D}} + \lambda I)^{-1}}. \tag{61}$$

Putting (58) and (61) together, we have

$$\textbf{Term (i)} \leq \mathcal{O}\left( \left[ \frac{3 + \exp(C)}{\sqrt{N}} \left( \sqrt{d + \log(1/\delta)} + \frac{\alpha}{\sqrt{N}} \kappa_{\mathcal{D}} \right) + \sqrt{\lambda C^2} \right] \cdot \left\| \mathop{\mathbb{E}}_{\substack{x \sim \rho, \\ y \sim \pi^\star(\cdot|x)}} [\phi(x,y)] \right\|_{(\Sigma_{\mathcal{D}} + \lambda I)^{-1}} \right). \tag{62}$$

**Step 3: bounding term (ii).** We can decompose and bound term (ii) by

$$J(\widehat{r}, \widehat{\pi}) - J(r^\star, \widehat{\pi}) = J(\widehat{r}, \widehat{\pi}) + \frac{1}{\alpha} \ell(\widehat{r}, \mathcal{D}) - \left( J(r^\star, \widehat{\pi}) + \frac{1}{\alpha} \ell(r^\star, \mathcal{D}) \right) + \frac{1}{\alpha} (\ell(\widehat{r}, \mathcal{D}) - \ell(r^\star, \mathcal{D}))$$

$$\overset{(i)}{\leq} \frac{1}{\alpha} (\ell(\widehat{r}, \mathcal{D}) - \ell(r^\star, \mathcal{D}))$$

$$\leq \frac{1}{\alpha} (\ell(\widehat{r}, \mathcal{D}) - \ell(r_{\mathsf{MLE}}, \mathcal{D}) + \ell(r_{\mathsf{MLE}}, \mathcal{D}) - \ell(r^\star, \mathcal{D})),$$

where (i) follows from the fact that $(\widehat{r}, \widehat{\pi})$ is a saddle point. Due to convexity of $\ell$, we have

$$\ell(\widehat{r}, \mathcal{D}) - \ell(r_{\mathsf{MLE}}, \mathcal{D}) \leq \left\langle \nabla_\theta \ell(\widehat{r}, \mathcal{D}), \widehat{\theta} - \theta_{\mathsf{MLE}} \right\rangle$$

$$= \left\langle -\alpha \mathop{\mathbb{E}}_{\substack{x \sim \rho, \\ y \sim \widehat{\pi}(\cdot|x)}} [\phi(x,y)] - \lambda_1 \mathop{\mathbb{E}}_{\substack{x \sim \rho, \\ y \sim \pi_{\mathrm{cal}}(\cdot|x)}} [\phi(x,y)], \widehat{\theta} - \theta_{\mathsf{MLE}} \right\rangle$$

$$= -\alpha \left\langle \mathop{\mathbb{E}}_{\substack{x \sim \rho, \\ y \sim \widehat{\pi}(\cdot|x)}} [\phi(x,y)] - \mathop{\mathbb{E}}_{\substack{x \sim \rho, \\ y \sim \pi_{\mathrm{cal}}(\cdot|x)}} [\phi(x,y)], \widehat{\theta} - \theta_{\mathsf{MLE}} \right\rangle$$

$$\leq \alpha \kappa_{\mathcal{D}} \| \widehat{\theta} - \theta_{\mathsf{MLE}} \|_{\Sigma_{\mathcal{D}} + \lambda I}$$

$$\leq \frac{\alpha^2 (3 + \exp(C))}{N} \kappa_{\mathcal{D}}^2 + 2\sqrt{\lambda C^2} \alpha \kappa_{\mathcal{D}},$$

where the last step is due to (60). On the other hand, with probability $1 - \delta$ we have (Zhan et al., 2023, Lemma 1):

$$\ell(r_{\mathsf{MLE}}, \mathcal{D}) - \ell(r^\star, \mathcal{D}) \leq \widetilde{\mathcal{O}}(1).$$

Putting pieces together,

$$\textbf{Term (ii)} \leq \frac{\alpha(3 + \exp(C))}{N} \kappa_{\mathcal{D}}^2 + 2\sqrt{\lambda C^2} \kappa_{\mathcal{D}} + \frac{1}{\alpha}. \tag{63}$$

**Step 4: putting things together.** Combining (55) (62), (63), with probability $1 - \delta$ we have

$$J^\star(r^\star) - J(r^\star, \widehat{\pi})$$

$$\leq \mathcal{O}\left( \frac{1}{\sqrt{N}} \left[ (3 + \exp(C)) \left( \sqrt{d + \log(1/\delta)} + \kappa_{\mathcal{D}} \right) + C \right] \cdot \left\| \mathop{\mathbb{E}}_{\substack{x \sim \rho, \\ y \sim \pi^\star(\cdot|x)}} [\phi(x,y)] \right\|_{(\Sigma_{\mathcal{D}} + \lambda I)^{-1}} \right.$$

$$\left. + \frac{1}{\sqrt{N}} \left( (3 + \exp(C)) \kappa_{\mathcal{D}}^2 + 2C\kappa_{\mathcal{D}} + 1 \right) \right).$$

Here we have set $\alpha = \sqrt{N}$ and $\lambda = 1/N$. We conclude by bounding $\kappa_{\mathcal{D}}$ as

$$\kappa_{\mathcal{D}}^2 = \left\| \mathop{\mathbb{E}}_{\substack{x \sim \rho, \\ y \sim \widehat{\pi}(\cdot|x)}} [\phi(x,y)] - \mathop{\mathbb{E}}_{\substack{x \sim \rho, \\ y \sim \pi_{\mathrm{cal}}(\cdot|x)}} [\phi(x,y)] \right\|_{(\Sigma_{\mathcal{D}} + \lambda I)^{-1}}^2$$

$$\leq \left\| \mathop{\mathbb{E}}_{\substack{x \sim \rho, \\ y \sim \widehat{\pi}(\cdot|x)}} [\phi(x,y)] - \mathop{\mathbb{E}}_{\substack{x \sim \rho, \\ y \sim \pi_{\mathrm{cal}}(\cdot|x)}} [\phi(x,y)] \right\|_2^2 \cdot \left\| (\Sigma_{\mathcal{D}} + \lambda I)^{-1} \right\|_2$$

$$\leq 4(\lambda_{\min}(\Sigma_{\mathcal{D}}) + \lambda)^{-1}.$$

## D    EXPERIMENTAL DETAILS

### D.1    RLHF FOR LLMS — OFFLINE SETTING

For the offline setting experiments, we adopt instruction tuned models, LLAMA2-7B-CHAT, LLAMA2-13B-CHAT and FLAN-T5-XL as the base models. To prompt these models, we prepend the questions in `ARC-Challenge` task (Clark et al., 2018) with

*What is the choice to the following Question? Only provide the choice by providing a single letter.*

and further append the question with

*The answer is:.*

The question is structured in a way that the multiple choices are appended with alphabets (letters) within parenthesis to the question. As an example:

*Question: George wants to warm his hands quickly by rubbing them. Which skin surface will produce the most heat? Choices: (A)dry palms (B)wet palms (C)palms covered with oil (D)palms covered with lotion*

We set $\pi_{\text{cal}}$ to the empirical distribution of the ground truth answer which is known to us. Based on preliminary experiments, we set $\beta$ as 0.1 in DPO and $\tau$ as 1.0 in IPO. For VPO, we experiment with moving $\alpha$ from 0.01 to 10, choosing 1 for the reported results. For all models, we train the base models with different algorithms DPO, VPO and IPO for 3000 steps and report the accuracy of the performance on the `ARC-challenge` test data set after every 500 steps. The training for LLAMA2-13B-CHAT model on 128 TPU-v4 takes around 2hrs and for FLAN-T5-XL on 64 TPU-v3 takes 1 hour.

### D.2    RLHF FOR LLMS — ONLINE SETTING

#### D.2.1    BUFFER

The prompt used for generating AI feedback (and rating) for `TL;DR` summarization is identical to (Guo et al., 2024). We set $\pi_{\text{cal}}$ to the empirical distribution of the negative answer pairs $(x, y_-)$ collected by the policy. We set $\beta$ as 0.1 for the DPO term similar to (Guo et al., 2024). Additionally for VPO, we decrease the coefficient exponentially following $\frac{\alpha}{\sqrt{1+\text{training steps}}}$. We try different values of $\alpha$ and report the results for 0.1 and 0.01.

The training of the policy, PALM2-XXS on 64 TPU-v3 for 5000 steps takes around 12 hours for both online DPO and VPO. We report the win rate percentage against the base SFT model for every 1000 steps using PALM2-XS judge. We also further conduct side by side comparison of Online DPO and VPO at 5000 step.

#### D.2.2    ITERATIVE

The UltraFeedback data (Cui et al., 2023) contains around 61k preference pairs of single-turn conversations. We divide this data set into 4 chunks. We use the first chunk to train a DPO model which we refer to as Zephyr-7B-DPO in Table 1. We use the remaining 3 chunks to train consecutive iterations of DPO and VPO, using the checkpoint from the previous iteration to initialize the policy for the current iteration. For each iteration, the prompts from the data are extracted and a new answer is sampled from the policy. We label the data using online AI feedback provided by PairRM (Jiang et al., 2023), using the similar ranking procedure as (Zhang et al., 2024b) where the new sample is ranked against $y_w$ and $y_l$ from the data. For VPO, we also adopt another ranking process where we sample two answer from the policy which are ranked against each other. We also set $\pi_{\text{cal}}$ to the empirical distribution of the negative answer pairs $(x, y_-)$ collected by the policy. We report the results of the best checkpoint of these two ranking procedures in Table 1. All experiments are conducted on 16xA100 GPUs.

| Model | AlpacaEval 2.0 | | | MT-Bench | | |
|---|---|---|---|---|---|---|
| | LC Win Rate | Win Rate | Avg. Len | Avg | 1st Turn | 2nd Turn |
| Zephyr-7B-SFT | 8.01 | 4.63 | 916 | 5.30 | 5.63 | 4.97 |
| Zephyr-7B-DPO | 15.41 | 14.44 | 1752 | 7.31 | 7.55 | 7.07 |
| DPO Iter 1 (Zephyr) | 20.53 | 16.69 | 1598 | 7.53 | 7.81 | 7.25 |
| DPO Iter 2 (Zephyr) | 22.12 | 19.82 | 1717 | 7.55 | 7.85 | 7.24 |
| DPO Iter 3 (Zephyr) | 22.19 | **19.88** | 1717 | 7.46 | 7.85 | 7.06 |
| VPO Iter 1 (Zephyr) | **22.53** | 19.09 | 1638 | 7.50 | 7.76 | 7.24 |
| VPO Iter 2 (Zephyr) | 21.84 | 18.78 | 1663 | **7.62** | 7.93 | **7.32** |
| VPO Iter 3 (Zephyr) | 22.15 | 19.58 | 1713 | 7.61 | **8.01** | 7.21 |
| SPIN | 7.23 | 6.54 | 1426 | 6.54 | 6.94 | 6.14 |
| Orca-2.5-SFT | 10.76 | 6.99 | 1174 | 6.88 | 7.72 | 6.02 |
| DNO (Orca-2.5-SFT) | 22.59 | 24.97 | 2228 | 7.48 | 7.62 | 7.35 |
| Mistral-7B-Instruct-v0.2 | 19.39 | 15.75 | 1565 | 7.51 | 7.78 | 7.25 |
| SPPO (Mistral-it) | 28.53 | 31.02 | 2163 | 7.59 | 7.84 | 7.34 |
| Yi-34B-Chat | 27.19 | 21.23 | 2123 | 7.90 | - | - |
| Llama-3-70B-Instruct | 33.17 | 33.18 | 1919 | 9.01 | 9.21 | 8.80 |
| GPT-4 Turbo (04/09) | 55.02 | 46.12 | 1802 | 9.19 | 9.38 | 9.00 |

Table 1: Results on AlpacaEval 2.0 and MT-Bench.

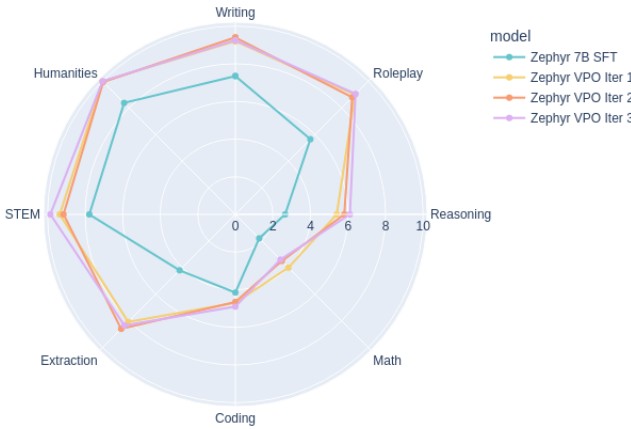

Figure 3: Radar chart of MT-Bench results for Zephyr-7B.

### D.3 SYNTHETIC BANDIT PROBLEMS

We evaluate the proposed methods on two synthetic problems: i) a multi-armed bandit problem, and ii) a linear contextual bandit problem.

**Multi-armed bandit (MAB) problem.** In this scenario, we set $|\mathcal{X}| = 1$ and $|\mathcal{Y}| = 10$. For each $y \in \mathcal{Y}$, the ground truth reward $r^\star(x, y)$ is randomly generated i.i.d. from a uniform distribution $U([0, 1])$. The policy is parameterized by $\pi_\theta(\cdot|x) = \mathsf{softmax}(\theta(x, \cdot))$, where $\theta \in \mathbb{R}^{10}$. The reference policy $\pi_{\mathrm{ref}}$ is set to $\pi_{\theta_{\mathrm{ref}}}$ with $\theta_{\mathrm{ref}}(x, y)$ sampled i.i.d. from $U([0, 1])$.

**Linear contextual bandit problem.** Here, we set $\mathcal{X} = \mathbb{R}^2$, $|\mathcal{Y}| = 50$. For each $(x, y)$ pair, the ground truth reward is given by $r^\star(x, y) = \langle \phi(x, y), \theta^\star \rangle$, where $\theta^\star \in \mathbb{R}^{10}$ is randomly sampled from $U([0, 1])$, and the feature vector $\phi(x, y)$ is the output of the hidden layer of a fixed two-layer MLP, with the input given by the concatenation of $x$ and the one-hot encoding of $y$. The activation function is set to tanh. The context vector $x$ is drawn from standard normal distribution. We focus on log-linear policy class $\pi_\theta(\cdot|x) = \mathsf{softmax}(\langle \theta, \phi(x, \cdot) \rangle)$, and set $\pi_{\mathrm{ref}} = \pi_{\theta_{\mathrm{ref}}}$ with $\theta_{\mathrm{ref}}(x, y)$ sampled i.i.d. from $U([0, 1])$.

For both problem we set $\pi_{\mathrm{ref}} = \pi_{\mathrm{b}} = \pi_{\mathrm{cal}}$ and use mini-batch sample of size 5 in every iteration. We approximately solve the optimization problems by performing 20 AdamW optimization steps with learning rate 0.01 and weight decay rate 0.01 in every iteration for the online setting and 1000 steps

for the offline setting. We set $\beta = 5$ for the online linear contextual bandit problem and $\beta = 1$ for all other experiments to better illustrate the performance differences.

We plot the average results over 50 independent runs for both experiments in Figure 4 and Figure 5. As demonstrated in Figure 4, an appropriate choice of $\alpha$ allows our method to outperform the model-based MAB with MLE baseline in the long-term performance of cumulative regret, at the cost of slightly increased cumulative regret in the first 100 iterations. This highlights the effectiveness of the VPO in achieving more principled exploration-exploitation trade-off. For the offline setting, Figure 5 demonstrates that the performance of both MLE-MAB and VPO improves as the number of offline data increases. However, VPO achieves a consistently lower sub-optimality gap compared with that of MLE-MAB.

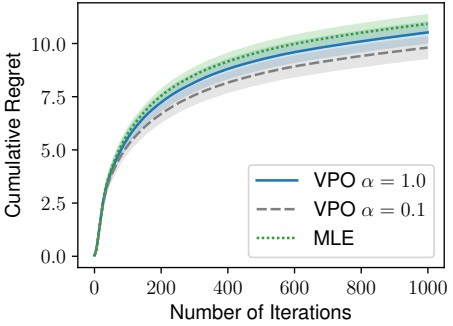 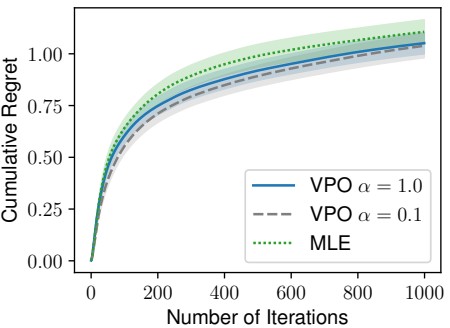

Figure 4: The cumulative regret v.s. number of iterations plot of VPO and MLE-MAB methods in the online MAB problem (left panel) and online linear contextual bandit problem (right panel), respectively.

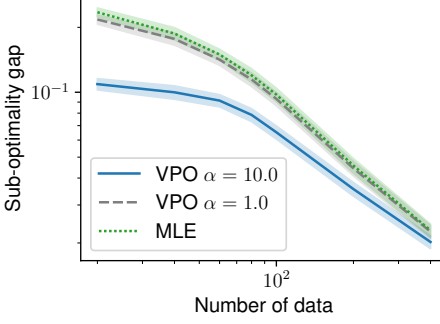 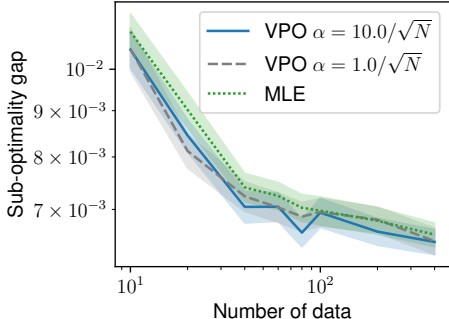

Figure 5: The sub-optimality gap v.s. number of data plot of VPO and MLE-MAB methods in the offline MAB problem (left panel) and offline linear contextual bandit problem (right panel), respectively.

