# OpenReview forum: "Value-Incentivized Preference Optimization: A Unified Approach to Online and Offline RLHF"
_ICLR.cc/2025/Conference — ICLR 2025 Poster_

### Official Review · Reviewer_S2VA · 2024-10-29

**Soundness:** 3
**Presentation:** 3
**Contribution:** 3
**Rating:** 6
**Confidence:** 4

**Summary:**

This paper introduces Value-Incentivized Preference Optimization (VPO), a novel method designed to unify online and offline Reinforcement Learning from Human Feedback (RLHF). The primary goal of VPO is to address the challenge of incorporating uncertainty estimation into the reward function, a key issue in both online and offline RLHF. VPO regularizes the maximum likelihood estimate of the reward function using the value function, modulated by a sign to indicate whether optimism or pessimism is chosen. The method also directly optimizes the policy with implicit reward modeling, simplifying the RLHF pipeline. Theoretical guarantees for VPO are provided, showing that it matches the performance rates of standard RL algorithms in both online and offline settings. Experimental results on text summarization, dialogue, and standard benchmarks verify the practicality and effectiveness of VPO.

**Strengths:**

1. VPO provides a unified approach to both online and offline RLHF, addressing a significant gap in the literature. This makes it applicable to a wide range of scenarios where preference data is either abundant or scarce.

2. The paper offers strong theoretical guarantees, demonstrating that VPO matches the performance rates of standard RL algorithms. This theoretical foundation adds credibility to the practical applicability of VPO.

3. VPO is designed to be practically implementable and theoretically grounded, making it suitable for large language models (LLMs). The regularization technique using the value function is straightforward and computationally feasible.

4. By directly optimizing the policy with implicit reward modeling, VPO simplifies the RLHF pipeline. This reduces the complexity and computational cost of training LLMs, making the method more accessible and efficient.

5. The paper includes comprehensive experiments on various tasks such as text summarization, dialogue, and standard benchmarks. These experiments demonstrate that VPO is effective and practical, outperforming or matching existing methods in different scenarios.

6. The work suggests a broader methodology for designing practical algorithms with principled optimism or pessimism under more general RL setups, opening avenues for future research and applications.

**Weaknesses:**

1. Although the paper covers several tasks, the scope of the experiments might be limited. More diverse and challenging tasks, particularly those involving real-world applications, could further validate the robustness and generalizability of VPO. While the paper claims that VPO matches or outperforms existing methods, a more detailed comparison with state-of-the-art techniques, including recent advancements in RLHF, would strengthen the claims and provide a clearer picture of VPO's advantages.

2. The theoretical guarantees provided are based on certain assumptions, such as well-behaved reward functions and policies. These assumptions may not always hold in real-world scenarios, limiting the applicability of VPO in more complex environments.

3. The paper does not extensively discuss the scalability of VPO to extremely large datasets or models. Addressing these scalability issues would be crucial for deploying VPO in industrial-scale applications.

**Questions:**

Is VPO equal to the DPO+KL penalty?

---

> ### Author Response · Authors · 2024-11-23
>
> Thanks for your time reviewing our paper and for your feedback.
>
> >**(W1) Although the paper covers several tasks, the scope of the experiments might be limited. More diverse and challenging tasks, particularly those involving real-world applications, could further validate the robustness and generalizability of VPO. While the paper claims that VPO matches or outperforms existing methods, a more detailed comparison with state-of-the-art techniques, including recent advancements in RLHF, would strengthen the claims and provide a clearer picture of VPO's advantages.**
>
> Thank you for the valuable feedback. The main contribution of the paper is not to try to provide SOTA numbers on all the benchmarks, but introducing the value-inceentived regularization as a theoretically sound scheme that can be flexibly incorporated into the practical pipeline. We believe the main contribution of this paper is on the theoretical front. Nonetheless, we thrive to provide additonal numerical experiments to validate our proposed approach for different benchmarks and LLM models, following the suggestion. We provide the additional early promising numerical experiments in a separate global response.
>
> >**(W2) The theoretical guarantees provided are based on certain assumptions, such as well-behaved reward functions and policies. These assumptions may not always hold in real-world scenarios, limiting the applicability of VPO in more complex environments.**
> >
> We acknowledge your point about the assumptions in our theoretical analysis. However, we would like to clarify several aspects:
>
> - The assumptions we make (like linear reward functions) are primarily for analytical tractability and to clearly demonstrate the core principles of VPO. Our analysis could be naturally extended to more general reward function classes using existing techniques from the literature, such as those involving neural network function approximation with appropriate covering number arguments, as demonstrated in [1][2].
>
> - The assumptions we make are standard in the RLHF literature and not particularly restrictive. Similar assumptions appear in several foundational works such as [3],[4].
>
> > **(W3) The paper does not extensively discuss the scalability of VPO to extremely large datasets or models. Addressing these scalability issues would be crucial for deploying VPO in industrial-scale applications.**
>
> VPO is designed to be highly scalable, inheriting the same computational efficiency that has made DPO successful in industrial applications. Like DPO, VPO directly optimizes the policy while avoiding explicit reward modeling, maintaining a simple and efficient RLHF pipeline. The key difference is the addition of a value-incentivized regularization term, which introduces minimal computational overhead. Our empirical results also demonstrate this scalability. For example, we successfully applied VPO to large-scale models like LLAMA2-13B.
>
> > **Q: Is VPO equal to the DPO+KL penalty?**
>
> The value-incentivized regularization term in VPO can be written in the form of a reversed KL divergence, withiout impacting the optimization (see our discussion in line 262-269). However, VPO's formulation offers additional flexibility by designing the calibration policy.
>
>
> ---
>
> [1] Liu, Z. et al., 2024. Maximize to explore: One objective function fusing estimation, planning, and exploration.
>
> [2] Jin C. et al., 2022. The power of exploiter: Provable multi-agent RL in large state spaces.
>
> [3] Zhu B. et al., 2023. Principled reinforcement learning with human feedback from pairwise or k-wise comparisons.
>
> [4] Zhan W. et al., 2023. Provable Offline Preference-Based Reinforcement Learning.

---

> > ### Comment · Reviewer_S2VA · 2024-11-27
> > **Thanks**
> >
> > Thank you for your response, which has addressed most of my concerns. I believe this work has certain merits, but the idea of regularization is not sufficiently novel. Therefore, I will stick to my final score.

---

> > > ### Author Response · Authors · 2024-11-27
> > >
> > > We are happy to hear that our response has addressed most of your concerns! Please don't hesitate to single out your remaining ones, as we will be very happy to address them further.
> > >
> > > Regarding the regularization, we want to stress that the paper doesn't merely propose the idea of regularization, but rather, proposing the *right regularization term* to use that achieves simultaneous *theoretical* performance guarantees for both online and offline settings. A key novel strength of the paper is that, the proposed value-incentivized regularization term, although in the form of the optimal value of an inner value optimization problems, turns out can be reformulated in a rigorous manner that eliminates the cumbersome nested optimization. We believe this is of sufficient novelty and significance.

---

### Official Review · Reviewer_pr21 · 2024-10-30

**Soundness:** 3
**Presentation:** 3
**Contribution:** 2
**Rating:** 6
**Confidence:** 4

**Summary:**

The authors propose a new algorithm for RLHF, value-incentivized preference optimization (VPO), which allows for training an LLM with either optimistic or pessimistic principles in both offline and online settings. Experiments show VPO is more robust to over-optimization in the offline setting and outperforms online DPO in online settings.

**Strengths:**

- The theoretical motivation is clear and makes sense, and proposing both offline and online variants makes it more applicable in popular LLM training settings.
- The ARC-challenge experiments do seem to show that VPO does avoid over-optimization relative to DPO and IPO.

**Weaknesses:**

- The AlpacaEval results seem quite weak, with further iterations of VPO underperforming multiple iterations of DPO, and having a fairly small gap in the one iteration setting (2 points). It would be useful to either run more seeds or try more models (e.g., VPO on top of llama 3 models) to see if the improvement is robust.
- There has been a lot of work into DPO-like algorithms recently in the field (I think the recent rainbowPO paper [1] has a good discussion of them). It would be useful to discuss how the proposed offline algorithm relates to these other approaches, such as SimPO [2] or WPO [3] (which appear to perform much more strongly on benchmarks like AlpacaEval 2). I see there is some discussion about how the method relates to DPOP, which is good.
- Following the above, it would be good to see comparisons with these newer, reportedly better-performing DPO variants as well, to see if VPO still can perform better or similarly.

Overall, I think the proposed algorithm is reasonable and incorporating principles of pessimism and optimism into RLHF is interesting, but I think that the empirical evidence that this leads to strong performance is somewhat weak. While the robustness to over-optimization is good, I think there is relatively little evidence in the work as-is that the proposed approach leads to solid improvements.

[1] Zhao et al. 2024. RainbowPO: A Unified Framework for Combining Improvements in Preference Optimization. https://arxiv.org/abs/2410.04203.

[2] Meng et al. 2024. SimPO: Simple Preference Optimization with a Reference-Free Reward. https://arxiv.org/abs/2405.14734

[3] Zhou et al. 2024. WPO: Enhancing RLHF with Weighted Preference Optimization. https://arxiv.org/abs/2406.11827

**Questions:**

- How does the proposed approach differ to and work with other *PO approaches? (as mentioned in weaknesses)

---

> ### Author Response · Authors · 2024-11-23
>
> Thanks for your time reviewing our paper and for your feedback.
>
>
> >**(W2) There has been a lot of work into DPO-like algorithms recently in the field (I think the recent rainbowPO paper has a good discussion of them). It would be useful to discuss how the proposed offline algorithm relates to these other approaches, such as SimPO or WPO (which appear to perform much more strongly on benchmarks like AlpacaEval 2). I see there is some discussion about how the method relates to DPOP, which is good.**
>
> Thank you for the suggestion! In the revision, we have included a brief discussion on SimPO and WPO, highlighting that our proposed algorithm can be integrated with these approaches. We want to hightlight that our contribution is theoretically demonstrating the value-incentived regularization scheme enables provably efficient online/offline RLHF, which can be further cobined with other algorithmic schemes as mentioned above. We believe this theoretical contribution sets apart our approach from other existing literature.
>
> >**(W1) The AlpacaEval results seem quite weak, with further iterations of VPO underperforming multiple iterations of DPO, and having a fairly small gap in the one iteration setting (2 points). It would be useful to either run more seeds or try more models (e.g., VPO on top of llama 3 models) to see if the improvement is robust.**
>
> >**(W3) Following the above, it would be good to see comparisons with these newer, reportedly better-performing DPO variants as well, to see if VPO still can perform better or similarly.**
>
> Thank you for the valuable feedback. The main contribution of the paper is not to try to provide SOTA numbers on all the benchmarks, but introducing the value-inceentived regularization as a theoretically sound scheme that can be flexibly incorporated into the practical pipeline. We believe the main contribution of this paper is on the theoretical front. Nonetheless, we thrive to provide additonal numerical experiments to validate our proposed approach for different benchmarks and LLM models, following the suggestion. We provide the additional early promising numerical experiments in a separate global response.
>
>
> >**(Q1)How does the proposed approach differ to and work with other \*PO approaches? (as mentioned in weaknesses)**
>
> - SimPO is an empirical paper that proposes using length-normalized log probabilities as an implicit reward function and adds a target reward margin to improve instruction-following performance.
> - WPO is an empirical paper that proposes reweighting preference pairs according to their probability under the current policy to simulate on-policy learning with off-policy data.
> - In contrast, our work focuses on the theoretical foundations of incorporating uncertainty in RLHF through value-incentivized regularization, providing convergence guarantees for both online and offline settings. We remark that SimPO/WPO and VPO are complementary: the value-incentivized regularization term we derive could be naturally combined with SimPO/WPO's objective.

---

> > ### Comment · Reviewer_pr21 · 2024-11-24
> > **Re: response**
> >
> > Hi, thank you for your response! After some consideration, I've bumped my score up to 6. I think the experimental results are still a bit weak, but the theoretical contribution is useful and I agree that combining the ideas presented here with approaches like SimPO or WPO would be a interesting future direction.

---

### Official Review · Reviewer_Er6X · 2024-11-04

**Soundness:** 3
**Presentation:** 3
**Contribution:** 3
**Rating:** 5
**Confidence:** 3

**Summary:**

Reinforcement Learning from Human Feedback (RLHF) is emerged as one of the most powerful approach to finetune the large language model (LLM) and align its performance to the human. However, even with the reward model-less approach like DPO, the uncertainty of the preference label remains as a core challenge. This work, VPO, highly exploits the closed form formulation between the reward function and optimal policy. From the intuition of the policy-invariance against prompt-dependent reward function shifting, they derive a tractable objective function that reflects the pessimism / optimism of the uncertainty. Proposed objective function has a guarantee of the uncertainty bound, and experimental results show an outperforming output than the baseline.

**Strengths:**

Their formulation and statements, thesis, have solid foundation and concrete derivation. They provide a reasonable error bound that allows one to estimate the difference between the trained policy and the optimal policy. Their base framework is easily applicable to the both online and offline settings with minimal modification and its implementation is quite simple. The language of the paper is tidy and easy to read.

**Weaknesses:**

Above all, the empirical gap is insufficient. Especially on table 1, the performance gap is around 0.5%p, therefore proper statistical tests are required to verify the effectiveness of the proposed method. Furthermore, Iter 3 setting tends to underperform than Iter 1 and Iter 2, thus it diminishes the intuition of theorem 1. The second concern is the misleading introduction. To my best knowledge, the term 'uncertainty' in the RLHF context usually refers to the ambiguousity and noise of the preference label. However, due to my understanding, this paper mainly focuses on the uncertainty and errors that occurred from the imperfect optimization of the RL framework itself. This part might be rewritten to avoid misleading and highlight their contribution.

**Questions:**

Is there any statistical result to validate the significance of the performance gap in Table 1? The gap looks marginal without any additional information. Reporting the standard deviation may be helpful, in my opinion. Rewriting your focus will reduce the misleading; the term uncertainty usually sounds like an error from the noisy label. Finally, it may be a subjective comment; the weight of the theory content was relatively high, and it reduced the volume of the experimental contents. I wish some more ablation studies could support your core contribution to the guaranteed confidence boundary.

---

> ### Author Response · Authors · 2024-11-23
>
> Thanks for your time reviewing our paper and for your feedback.
>
> >**(W1) the empirical gap is insufficient. Especially on table 1, the performance gap is around 0.5%p, therefore proper statistical tests are required to verify the effectiveness of the proposed method. Furthermore, Iter 3 setting tends to underperform than Iter 1 and Iter 2, thus it diminishes the intuition of Theorem 1.**
>
> The observation that Iter 3 sometimes underperforms Iter 1 and 2 does not contradict Theorem 1, as the theorem establishes an upper bound on the cumulative regret rather than guaranteeing monotonic improvement. This non-monotonic behavior is actually a common phenomenon in iterative learning processes due to the stochastic nature of the training process and data. Similar non-monotonic behavior also occurs in DPO's performance (see Table 1).
>
> >**(W2) Misleading introduction: 'uncertainty' in the RLHF context usually refers to the ambiguousity and noise of the preference label...this paper mainly focuses on the uncertainty and errors that occurred from the imperfect optimization of the RL framework itself. This part might be rewritten to avoid misleading and highlight their contribution.**
>
>
> We would like to clarify that this work primarily focuses on model uncertainty (epistemic uncertainty) in reward learning rather than optimization error. This is fundamentally tied to the fact that we only observe pairwise preferences rather than direct reward values - the preference data could be seen as a noisy reward signal that requires careful uncertainty estimation.
>
> To understand why model uncertainty is crucial and unavoidable in this context, consider the Upper Confidence Bound (UCB) principle in reinforcement learning - where an agent maintains uncertainty estimates to balance exploration and exploitation. While this principle is well-established, implementing it for LLMs becomes computationally prohibitive because standard techniques for constructing confidence intervals become intractable under arbitrary policy parameterizations (as discussed in [1]).
>
> Rather than trying to explicitly estimate these uncertainties (which would be computationally infeasible), VPO provides a practical solution by regularizing the maximum-likelihood estimate of the reward function using the value function. This allows us to incorporate the benefits of uncertainty-aware learning without the computational burden of explicit uncertainty estimation. For the **first** time, we bypasses the fundamental difficulty of direct uncertainty quantification while still maintaining the benefits of uncertainty-aware learning in the RLHF pipeline.
>
>
>
> >**(Q1) Is there any statistical result to validate the significance of the performance gap in Table 1? The gap looks marginal without any additional information. Reporting the standard deviation may be helpful.**
>
> Thank you for your suggestion. Due to resource constraints, we report one-shot results for LLM training in Figure 1 and Table 1, which aligns with common practice in RLHF literature (e.g., [2][3]). However, we are actively conducting additional experiments in both online and offline settings, using various models, to provide a more comprehensive evaluation of VPO. We provide additional results in the seperate response summarizing the new experiments.
>
> >**(Q2) The weight of the theory content was relatively high, and it reduced the volume of the experimental contents. Some more ablation studies could support your core contribution to the guaranteed confidence boundary**
>
> Thanks for the suggestion! We conducted ablation studies in synthetic bandit problems by experimenting with different regularization coefficients $\alpha$, as illustrated in Figure 4 and 5.
>
> The theoretical emphasis in our paper is intentional and aligns with our primary goal: establishing a principled foundation for incorporating uncertainty in RLHF. While existing works have proposed various heuristic approaches, our key contribution is providing the first theoretically-grounded framework that unifies both online and offline RLHF settings with provable guarantees. The experimental results, though more concise, effectively validate our theoretical insights and demonstrate practical benefits like improved robustness against overoptimization. This theory-first approach has enabled us to develop a more principled solution to a fundamental challenge in RLHF.
>
> ---
>
> [1] Osband et al., 2023. Epistemic Neural Networks.
>
> [2] Chen et al., 2025. Self-play fine-tuning converts weak language models to strong language models.
>
> [3] Wu et al., 2024. Self-play preference optimization for language model alignment.

---

> ### Comment · Reviewer_Er6X · 2024-11-25
> **Comment**
>
> Thank you for your comment. I will maintain my score.

---

### Official Review · Reviewer_Rtts · 2024-11-04

**Soundness:** 2
**Presentation:** 3
**Contribution:** 3
**Rating:** 5
**Confidence:** 3

**Summary:**

The paper proposes VPO which can either optimistically or pessimistically compute the reward function so that it can be used for both online and offline settings. Interestingly, ignoring the prompt-dependent baseline reward value due to the BT model, there is a computationally efficient algorithm to compute the resulting policy. In fact the formulation results in adding a KL regularization to DPO.
The paper proved a regret bound for online setting and a PAC bound for offline setting.
The experimental results have several evidence to claim that VPO outperforms DPO.

**Strengths:**

The paper gives a new view of online/offline learning for RL research.

- Solving online and offline learning problems using the same algorithm with (theoretically) just an adjustment of one hyperparameter is very interesting.
- The method is simple and easy to implement.
- The assumption and the optimization criteria of the method is clearly stated so that what the method intended to optimize is clearly stated in the paper, which makes the understanding of the algorithm much better. It is connected to the body of RL research.
- The analytical results are nice to have, yet its practical implication is difficult to verify in experiments.

**Weaknesses:**

The impact of the paper to the field of natural language processing and large language model is not clear to me.

- In terms of practical benefit to the LLM development, the experimental results are not decisive to conclude that VPO is better than DPO. I believe that the number of runs is not reported. The standard deviation/error is also not reported (sorry if I missed it).
- From what I understand, ARC-challenge is not a benchmark for alignment. It is a multiple-question answering task just to evaluate the knowledge of the LLM. It is difficult to judge if VPO has an advantage over IPO/DPO for online learning settings given that aside from the format, ARC is mostly just a collection of discrete knowledge that shares little among each instance. I would like to take a look at the generation examples. Or, I would guess using the standard benchmarks for alignment algorithms like AlpacaFarm or HH-RLHF makes more sense.


----------------------------------
I believe my concern was clearly stated in my very first reviewer comment as above.

Why ARC.

The authors decided not to answer this question. So, there is no reason for me to consider improving my score.
They decided to separate my questions into multiple discrete questions so that it looked like I was just randomly asking for generation examples.

> > (Q2) I'm curious about why ARC-Challenge is chosen as the task for the offline learning task. ARC is a collection of questions evaluating the knowledge of the LLM. It is mostly used to evaluate the non-instruction tuned pretrained LLMs but not the result of the alignment process. Isn't the *POs just learning the format of the multiple question answering format?
>
> We would like to remark that Arc-challenge has been widely used in previous RLHF works including DPOP[1], and Iterative RPO [2]. We follow the protocol in [2], and side-by-side comparison to offline DPO and IPO demonstrates that pessimism is necessary.
>
> > I would like to take a look at the generation examples. Or, I would guess using the standard benchmarks for alignment algorithms like AlpacaFarm or HH-RLHF makes more sense.


I had to write multiple responses until the authors finally showed the result of the ARC at the last second of the discussion period.
I am confident that my question was clearly stated in my first official review and also stated repeatedly, so I believe it is not unfair to point out the following concern.

> {"inputs": {"inputs_pretokenized": "What is the choice to the following Question? Only provide the choice by providing a single letter.\n\nQuestion: Which of the following would you most likely use to study the growth of an insect over time? Choices: (A)a clock that measures seconds (B)a hand lens with a large lens (C)several insects that feed on other insects (D)a glass aquarium with food for the insect\n\nThe answer is:", "targets_pretokenized": "(D)a glass aquarium with food for the insect"}, "prediction": "(C) several insects that feed on other insects"}
>
> {"inputs": {"inputs_pretokenized": "What is the choice to the following Question? Only provide the choice by providing a single letter.\n\nQuestion: A pharmaceutical company published experimental data showing that a new medication improved cholesterol levels in the people who participated in the study. Which professional most likely did not contribute to developing this medication? Choices: (A)a chemist (B)a biologist (C)a geologist (D)a physician\n\nThe answer is:", "targets_pretokenized": "(C)a geologist"}, "prediction": "(B) a biologist"}
>
> {"inputs": {"inputs_pretokenized": "What is the choice to the following Question? Only provide the choice by providing a single letter.\n\nQuestion: Monica grows vegetable plants in her garden. The plants have holes in their leaves and look unhealthy. Which task should Monica do first to solve this problem? Choices: (A)Add fertilizer to the soil. (B)Water the garden more often. (C)Plant different vegetables in the garden. (D)Observe the plants to identify the source of the damage.\n\nThe answer is:", "targets_pretokenized": "(D)Observe the plants to identify the source of the damage."}, "prediction": "(C) Plant different vegetables in the garden."}
>
> {"inputs": {"inputs_pretokenized": "What is the choice to the following Question? Only provide the choice by providing a single letter.\n\nQuestion: The students in a class would like to make 20 paper sailboats for a race. The students will select one design and collect the materials they need to construct the boats. Which of the following is the best way for the students to be sure the paper sailboats will float without tipping over in the water? Choices: (A)construct a prototype of a boat for testing (B)calculate the total mass of all of the finished boats (C)determine the total amount of weight each boat can carry (D)test the strength of each material used to construct the boats\n\nThe answer is:", "targets_pretokenized": "(A)construct a prototype of a boat for testing"}, "prediction": "(D)test the strength of each material used to construct the boats."}
>
> {"inputs": {"inputs_pretokenized": "What is the choice to the following Question? Only provide the choice by providing a single letter.\n\nQuestion: A class tested the amount of vitamin C in fresh orange juice and calculated a different amount than the previous class that conducted the same investigation. What should most likely be done to make sure that their results are accurate? Choices: (A)try a different juice (B)analyze the recorded data (C)ask a scientist what went wrong (D)repeat the investigation two more times\n\nThe answer is:", "targets_pretokenized": "(D)repeat the investigation two more times"}, "prediction": "(B) analyze the recorded data"}.

Given that you are computing the accuracy using the exact match, "(A) XXX" is not a correct answer, even if the answer was "(A)XXX" because of the white space after the (B). 4 out of 5 generation examples insert the white space after the parenthesis in the base model. After the RLHF process, there seems to be no space after the parenthesis. It is likely to be a good reason why it has less accuracy than the RLHFed models.
This raises my concern that the whole RLHF process is just learning not to put a white space after the parenthesis. I don't call it an alignment.


The theory of the paper is interesting enough to make the paper accepted if accompanied by some experimental results to support the idea.
However, I don't think the experiments are conducted to evaluate what the algorithm is supposed to do.

------------------------------------------------------

Thank you very much for the extra experiments to verify the result on ARC.
I would put my score back to 5.

**Questions:**

- How many runs are conducted for Figure 1 and Table 1? Figures 4 and 5 show the average but do not show the standard deviation or standard error.
- I'm curious about why ARC-Challenge is chosen as the task for the offline learning task. ARC is a collection of questions evaluating the knowledge of the LLM. It is mostly used to evaluate the non-instruction tuned pretrained LLMs but not the result of the alignment process. Isn't the *POs just learning the format of the multiple question answering format? I also couldn't find how the evaluation is done for ARC. Do you consider only the exact match to be the correct answer?

---

> ### Author Response · Authors · 2024-11-23
>
> Thanks for your time reviewing our paper and for your feedback.
>
> > **(W1) In terms of practical benefit to the LLM development, the experimental results are not decisive to conclude that VPO is better than DPO. I believe that the number of runs is not reported. The standard deviation/error is also not reported.**
>
> >**(Q1) How many runs are conducted for Figure 1 and Table 1? Figures 4 and 5 show the average but do not show the standard deviation or standard error.**
>
>
> Thanks for the suggestion. We have updated Figure 4 and 5 to include standard error bars. We report one-shot results for LLM training in Figure 1 and Table 1 due to limited resources, as commonly performed in the RLHF literature, see, e.g., [3][4]. We are currently conducting more experiments on additional LLM models and benchmarks, and only able to share early promising results due to limited time. Please refer to the separate response on experiments. Nonetheless, we would like to highlight our main contribution is proposing a theoretically guaranteed approach to online/offline RLHF with rigorous regret guarantees, which are not available before.
>
>
> >**(W2) From what I understand, ARC-challenge is not a benchmark for alignment. It is a multiple-question answering task just to evaluate the knowledge of the LLM. It is difficult to judge if VPO has an advantage over IPO/DPO for online learning settings given that aside from the format, ARC is mostly just a collection of discrete knowledge that shares little among each instance.**
>
> >**(Q2) I'm curious about why ARC-Challenge is chosen as the task for the offline learning task. ARC is a collection of questions evaluating the knowledge of the LLM. It is mostly used to evaluate the non-instruction tuned pretrained LLMs but not the result of the alignment process. Isn't the \*POs just learning the format of the multiple question answering format?**
>
> We would like to remark that Arc-challenge has been widely used in previous RLHF works including DPOP[1], and Iterative RPO [2]. We follow the protocol in [2], and side-by-side comparison to offline DPO and IPO demonstrates that pessimism is necessary.
>
> >**I would like to take a look at the generation examples. Or, I would guess using the standard benchmarks for alignment algorithms like AlpacaFarm or HH-RLHF makes more sense.**
>
> We have provided online setting results with AlpacaEval 2.0 and MT-Bench in Table 1, which are also used in other RLHF works such as SPIN[3], SPPO[4] and DNO[5]. We did not use HH-RLHF because it is less challenging compared to the experiments we conduct. We are conducting further experiments on additional benchmarks, some early promising results are provided in the separate response summarizing the experiments.
>
>
> >**I also couldn't find how the evaluation is done for ARC. Do you consider only the exact match to be the correct answer?**
>
> Yes, we follow the setting in [1],[2] and consider only the exact match to be the correct answer.
>
> ---
>
> [1] Pal et al., 2024. Smaug: Fixing Failure Modes of Preference Optimisation with DPO-Positive.
>
> [2] Pang et al., 2024. Iterative Reasoning Preference Optimization.
>
> [3] Chen et al., 2025. Self-play fine-tuning converts weak language models to strong language models.
>
> [4] Wu et al., 2024. Self-play preference optimization for language model alignment.
>
> [5] Rosset et al., 2024. Direct nash optimization: Teaching language models to self-improve with general preferences.

---

> ### Comment · Reviewer_Rtts · 2024-11-25
>
> > Thanks for the suggestion. We have updated Figure 4 and 5 to include standard error bars.
>
> Thank you very much.
>
> > We report one-shot results for LLM training in Figure 1 and Table 1 due to limited resources, as commonly performed in the RLHF literature, see, e.g., [3][4].
>
> I do understand that it is commonly performed in RLHF literature and I understand that it is difficult or impossible to conduct large-scale experiments. However, I do not agree that such circumstances change the validity of scientific arguments.
>
> > Nonetheless, we would like to highlight our main contribution is proposing a theoretically guaranteed approach to online/offline RLHF with rigorous regret guarantees, which are not available before.
>
> Yes, I think the theoretical contribution of the paper is great. However, the paper is also advertised to have a practical implication, so I would also hope for it.
>
> > We would like to remark that Arc-challenge has been widely used in previous RLHF works including DPOP[1], and Iterative RPO [2].
>
> The existence of some papers using ARC for RLHF will not be a scientific reason to make it a good benchmark. I would like YOU to answer why ARC would be a suitable benchmark for RLHF. If it is explained in the prior RLHF papers using ARC, please cite it.
>
> > We have provided online setting results with AlpacaEval 2.0 and MT-Bench in Table 1, which are also used in other RLHF works such as SPIN[3], SPPO[4] and DNO[5].
>
> Thank you very much. It would be helpful if you can share the generation results of ARC, as I am very skeptical of it as an evaluation benchmark for RLHF.

---

> ### Author Response · Authors · 2024-11-25
> **generation results of ARC**
>
> Dear Reviewer Rtts,
>
> Here are some examples of inference on ARC-challenge. The inputs are questions and four options, and the prediction should be a single letter indicating the option.
>
> We would like to also thank the reviewer for recognizing our theoretical contribution. We believe it is greatly beneficial and desirable for the community to invest more on theoretically grounded RLHF methods. The experiments are used to demonstrate that our methods are useful in practice as well rather than to beat prior state-of-the-art.
>
> {"inputs": {"inputs_pretokenized": "What is the choice to the following Question? Only provide the choice by providing a single letter.\n\nQuestion: An astronomer observes that a planet rotates faster after a meteorite impact. Which is the most likely effect of this increase in rotation? Choices: (A) Planetary density will decrease. (B) Planetary years will become longer. (C) Planetary days will become shorter. (D) Planetary gravity will become stronger.\n\nThe answer is:", "targets_pretokenized": "C"}, "prediction": "C"}
>
> {"inputs": {"inputs_pretokenized": "What is the choice to the following Question? Only provide the choice by providing a single letter.\n\nQuestion: A group of engineers wanted to know how different building designs would respond during an earthquake. They made several models of buildings and tested each for its ability to withstand earthquake conditions. Which will most likely result from testing different building designs? Choices: (A) buildings will be built faster (B) buildings will be made safer (C) building designs will look nicer (D) building materials will be cheaper\n\nThe answer is:", "targets_pretokenized": "B"}, "prediction": "B"}
>
> {"inputs": {"inputs_pretokenized": "What is the choice to the following Question? Only provide the choice by providing a single letter.\n\nQuestion: The end result in the process of photosynthesis is the production of sugar and oxygen. Which step signals the beginning of photosynthesis? Choices: (A) Chemical energy is absorbed through the roots. (B) Light energy is converted to chemical energy. (C) Chlorophyll in the leaf captures light energy. (D) Sunlight is converted into chlorophyll.\n\nThe answer is:", "targets_pretokenized": "C"}, "prediction": "C"}
>
> {"inputs": {"inputs_pretokenized": "What is the choice to the following Question? Only provide the choice by providing a single letter.\n\nQuestion: A physicist wants to determine the speed a car must reach to jump over a ramp. The physicist conducts three trials. In trials two and three, the speed of the car is increased by 20 miles per hour. What is the physicist investigating when he changes the speed? Choices: (A) the control (B) the hypothesis statement (C) the dependent (responding) variable (D) the independent (manipulated) variable\n\nThe answer is:", "targets_pretokenized": "D"}, "prediction": "D"}
>
> {"inputs": {"inputs_pretokenized": "What is the choice to the following Question? Only provide the choice by providing a single letter.\n\nQuestion: An astronaut drops a 1.0 kg object and a 5.0 kg object on the Moon. Both objects fall a total distance of 2.0 m vertically. Which of the following best describes the objects after they have fallen a distance of 1.0 m? Choices: (A) They have each lost kinetic energy. (B) They have each gained the same amount of potential energy. (C) They have each lost the same amount of potential energy. (D) They have each gained one-half of their maximum kinetic energy.\n\nThe answer is:", "targets_pretokenized": "D"}, "prediction": "C"}

---

> > ### Comment · Reviewer_Rtts · 2024-11-26
> >
> > Thank you very much for sharing the generation results.
> >
> > Are these from the aligned model or the base model?
> > To clarify, the reason why I wish to see the generation results of ARC is that I doubt it to be a good benchmark for RLHF. To understand the behavior of the RLHF on the ARC benchmark, I would want to see the generation results from both before and after the RLHF procedure.

---

> ### Author Response · Authors · 2024-11-30
> **Examples of generation results for both before and after the RLHF procedure**
>
> Dear Reviewer Rtts,
>
> We attach generation results for both base model and aligned model as you requested (the results we posted above were from Zephry, and we reported using LLama2-7b for ARC-challenge in the paper; therefore it is a slip, sorry for the confusion).
>
> The base model is meta-llama/Llama-2-7b-chat-hf, and the following are results for 5 questions (for prefix and suffix of the prompts, please see Appendix D in the paper).
>
> {"inputs": {"inputs_pretokenized": "What is the choice to the following Question? Only provide the choice by providing a single letter.\n\nQuestion: Which of the following would you most likely use to study the growth of an insect over time? Choices: (A)a clock that measures seconds (B)a hand lens with a large lens (C)several insects that feed on other insects (D)a glass aquarium with food for the insect\n\nThe answer is:", "targets_pretokenized": "(D)a glass aquarium with food for the insect"}, "prediction": "(C) several insects that feed on other insects"}
>
> {"inputs": {"inputs_pretokenized": "What is the choice to the following Question? Only provide the choice by providing a single letter.\n\nQuestion: A pharmaceutical company published experimental data showing that a new medication improved cholesterol levels in the people who participated in the study. Which professional most likely did not contribute to developing this medication? Choices: (A)a chemist (B)a biologist (C)a geologist (D)a physician\n\nThe answer is:", "targets_pretokenized": "(C)a geologist"}, "prediction": "(B) a biologist"}
>
> {"inputs": {"inputs_pretokenized": "What is the choice to the following Question? Only provide the choice by providing a single letter.\n\nQuestion: Monica grows vegetable plants in her garden. The plants have holes in their leaves and look unhealthy. Which task should Monica do first to solve this problem? Choices: (A)Add fertilizer to the soil. (B)Water the garden more often. (C)Plant different vegetables in the garden. (D)Observe the plants to identify the source of the damage.\n\nThe answer is:", "targets_pretokenized": "(D)Observe the plants to identify the source of the damage."}, "prediction": "(C) Plant different vegetables in the garden."}
>
> {"inputs": {"inputs_pretokenized": "What is the choice to the following Question? Only provide the choice by providing a single letter.\n\nQuestion: The students in a class would like to make 20 paper sailboats for a race. The students will select one design and collect the materials they need to construct the boats. Which of the following is the best way for the students to be sure the paper sailboats will float without tipping over in the water? Choices: (A)construct a prototype of a boat for testing (B)calculate the total mass of all of the finished boats (C)determine the total amount of weight each boat can carry (D)test the strength of each material used to construct the boats\n\nThe answer is:", "targets_pretokenized": "(A)construct a prototype of a boat for testing"}, "prediction": "(D)test the strength of each material used to construct the boats."}
>
> {"inputs": {"inputs_pretokenized": "What is the choice to the following Question? Only provide the choice by providing a single letter.\n\nQuestion: A class tested the amount of vitamin C in fresh orange juice and calculated a different amount than the previous class that conducted the same investigation. What should most likely be done to make sure that their results are accurate? Choices: (A)try a different juice (B)analyze the recorded data (C)ask a scientist what went wrong (D)repeat the investigation two more times\n\nThe answer is:", "targets_pretokenized": "(D)repeat the investigation two more times"}, "prediction": "(B) analyze the recorded data"}.

---

> ### Author Response · Authors · 2024-11-30
> **Examples of generation results for both before and after the RLHF procedure con'd**
>
> And the following are results for the same 5 questions, but after RLHF procedure, using VPO.
>
> {"inputs": {"inputs_pretokenized": "What is the choice to the following Question? Only provide the choice by providing a single letter.\n\nQuestion: Which of the following would you most likely use to study the growth of an insect over time? Choices: (A)a clock that measures seconds (B)a hand lens with a large lens (C)several insects that feed on other insects (D)a glass aquarium with food for the insect\n\nThe answer is:", "targets_pretokenized": "(D)a glass aquarium with food for the insect"}, "prediction": "(D)a glass aquarium with food for the insect"}
>
> {"inputs": {"inputs_pretokenized": "What is the choice to the following Question? Only provide the choice by providing a single letter.\n\nQuestion: A pharmaceutical company published experimental data showing that a new medication improved cholesterol levels in the people who participated in the study. Which professional most likely did not contribute to developing this medication? Choices: (A)a chemist (B)a biologist (C)a geologist (D)a physician\n\nThe answer is:", "targets_pretokenized": "(C)a geologist"}, "prediction": "(C)a geologist"}
>
> {"inputs": {"inputs_pretokenized": "What is the choice to the following Question? Only provide the choice by providing a single letter.\n\nQuestion: Monica grows vegetable plants in her garden. The plants have holes in their leaves and look unhealthy. Which task should Monica do first to solve this problem? Choices: (A)Add fertilizer to the soil. (B)Water the garden more often. (C)Plant different vegetables in the garden. (D)Observe the plants to identify the source of the damage.\n\nThe answer is:", "targets_pretokenized": "(D)Observe the plants to identify the source of the damage."}, "prediction": "(D)Observe the plants to identify the source of the damage."}
>
>
> {"inputs": {"inputs_pretokenized": "What is the choice to the following Question? Only provide the choice by providing a single letter.\n\nQuestion: The students in a class would like to make 20 paper sailboats for a race. The students will select one design and collect the materials they need to construct the boats. Which of the following is the best way for the students to be sure the paper sailboats will float without tipping over in the water? Choices: (A)construct a prototype of a boat for testing (B)calculate the total mass of all of the finished boats (C)determine the total amount of weight each boat can carry (D)test the strength of each material used to construct the boats\n\nThe answer is:", "targets_pretokenized": "(A)construct a prototype of a boat for testing"}, "prediction": "(A)construct a prototype of a boat for testing"}
>
> {"inputs": {"inputs_pretokenized": "What is the choice to the following Question? Only provide the choice by providing a single letter.\n\nQuestion: A class tested the amount of vitamin C in fresh orange juice and calculated a different amount than the previous class that conducted the same investigation. What should most likely be done to make sure that their results are accurate? Choices: (A)try a different juice (B)analyze the recorded data (C)ask a scientist what went wrong (D)repeat the investigation two more times\n\nThe answer is:", "targets_pretokenized": "(D)repeat the investigation two more times"}, "prediction": "(D)repeat the investigation two more times"}

---

> ### Author Response · Authors · 2024-12-02
> **Clarification for misunderstandings/miscommunications**
>
> Dear Reviewer Rtts,
>
> We noticed that you lowered your rating, and after reviewing the matter, we believe this may be due to a misunderstanding or miscommunication. We would like to provide some clarification.
>
> First, we apologize for the delay in our feedback during the Thanksgiving holiday. Please know that we did not intend to reply at the last minute, and we hope this situation can be understood in the context of the holiday. Thank you for your patience, and we sincerely apologize for any confusion.
>
> Second, we reviewed the accuracy of the outputs for both the base model (Llama2-7b-chat) and the trained model. We performed post-processing on the answers and predictions, ensuring that we only counted a correct answer when the selected option was the target option. This means the accuracy is based on a match with the target option, not an exact string match. For the base model (Llama2-7b-chat), the accuracy is 606/1172 = 51.7%. For one checkpoint of the aligned models, the accuracy is 700/1172 = 59.7%. Therefore, the improvement in accuracy is not due to removing white space, as might have been misunderstood.
>
> We would like to also stress the point that ARC-challenge is just one task we used, and we do have other experiments (including the standard Alpaca and MT-bench) to corroborate the effectiveness of the proposed methods (which makes it more unlikely that the improvement is from some simple modifications of outputs).
>
> We hope this clears up any misunderstandings and helps in reconsidering the rating.
>
> Thank you again for your time and attention.
>
> Sincerely,
>
> The Authors

---

### Author Response · Authors · 2024-11-23
**Additional experiments and comments**

We thank all reviewers for their feedback. Following the suggestions, we significantly expanded our experimental evaluation:

- For the synthetic multi-armed bandits experiments, we have increased the number of runs and updated Figure 4 and 5 to include standard error bars.

- We are conducting more comprehensive experiments on additional LLM models and benchmarks. Due to limited time and resources, we finished part of the experiments on MT-bench, which show promising evidences for our proposed method. We will add the results for Alpaca Eval once they are finished.

1. VPO Zephyr `offline` version

|          | MT-Bench| | |
| -------- | ------- | -------- |-------- |
| | 1st Turn | 2nd Turn | Avg |
| Zephyr-7B-SFT  | 5.30 | 5.63 | 4.97 |
| Zephyr-7B-DPO-Iter1  | 7.53 | 7.81 | 7.25 |
| Zephyr-7B-DPO-Iter2  | 7.55 | 7.85 | 7.24 |
| Zephyr-7B-DPO-Iter3  | 7.46 | 7.85 | 7.06 |
| Zephyr-7B-Optimistic-VPO-Iter1  | 7.50 | 7.76 | 7.24 |
| Zephyr-7B-Optimistic-VPO-Iter2  | 7.62 | 7.93 | 7.32 |
| Zephyr-7B-Optimistic-VPO-Iter3  | 7.61 | 8.01 | 7.21 |
| Zephyr-7B-Pessimistic-VPO-Iter1  | 7.57 | 7.71 | **7.42** |
| Zephyr-7B-Pessimistic-VPO-Iter2  | 7.43 | 7.69 | 7.18 |
| Zephyr-7B-Pessimistic-VPO-Iter3  | 7.59 | 7.86 | 7.32 |

2. VPO using llama3-8b-instruct `online` and `offline`

|          | MT-Bench| | |
| -------- | ------- | -------- |-------- |
| | 1st Turn | 2nd Turn | Avg |
| Llama-3-8B-SFT  | - | - | 7.69 |
| Llama-3-8B-DPO-Iter1  | 8.41 | 7.55 | 7.98 |
| Llama-3-8B-DPO-Iter2  | 8.25 | 7.69 | 7.97 |
| Llama-3-8B-DPO-Iter3  | 8.13 | 7.62 | 7.88 |
| Llama-3-8B-Optimistic-VPO-Iter1  | 8.29 | 7.76 | **8.03** |
| Llama-3-8B-Optimistic-VPO-Iter2  | 8.27 | 7.59 | 7.94 |
| Llama-3-8B-Optimistic-VPO-Iter3  | 8.02 | 7.78 | 7.90 |
| Llama-3-8B-Pessimistic-VPO-Iter1  | 8.13 | 7.78 | 7.95 |
| Llama-3-8B-Pessimistic-VPO-Iter2  | 8.25 | 7.76 | 8.01 |
| Llama-3-8B-Pessimistic-VPO-Iter3  | 8.06 | 7.60 | 7.83 |


The improvement in our proposed VPO all comes from a simple (principled) mechanism (namely, the value-incentivized regularization), without adding other engineering tricks. Comparing with other concurrent works (such as RPO https://arxiv.org/pdf/2405.16436, and SimPO https://arxiv.org/pdf/2405.14734), it is fair to claim that the improvements over DPO on MT-Bench for our proposed methods are significant.

Moreover, we would like to emphasize that our main contribution is a theoretically-guaranteed algorithm (VPO) for online and offline RLHF. VPO, by introducing a simple value-incentivized regularization, provides rigorous theoretical guarantees for both online and offline settings while remaining practically implementable for LLMs. Our experimental results are strong compared to other theoretical RLHF works in the field. We didn't try to incorporate, and compete with, empirical tricks such as length regularization, but rather, aim to illustrate the value of the key idea proposed in this paper in the experiments.

In conclusion, our work makes a significant contribution by bridging the gap between theory and practice in RLHF, providing both strong theoretical guarantees and competitive empirical results on real-world tasks.

---

### Meta-Review · Area_Chair_Hmbu · 2024-12-22

**Metareview:**

This paper studied RLHF with pessimism and optimism, and both were achieved by a fairly uniform framework (all one needs is to flip the sign of the regularization). The proposed approach is theoretically grounded, and the algorithm is simple. The authors demonstrated their algorithm on multiple benchmarks. The weakness of the paper was the insufficient experiment. However, the authors added additional experiments during the rebuttal time, which addressed some reviewers' concerns.

**Additional Comments On Reviewer Discussion:**

The reviewers all acknowledged the authors' efforts during the rebuttal phase. The initial concerns from the negative reviews were around the experiments. There might be some misunderstanding between the authors and one of the reviewers initially, but at the end the reviewer acknowledged that the authors have improved the evaluation protocol of ARC. The authors also added additional experiments on MT-bench, which showed promising results. Overall, while the paper is on the borderline, it does have solid contribution from a theoretical perspective and it also demonstrated promising experiment results. We encourage the authors to include all these additional experiment results into the future versions of the paper.

---

### Decision · Program_Chairs · 2025-01-22

Accept (Poster)